# ConvT3: Structured State Kernels for Convolutional State Space Models

**Jaeyoung Hong**[*,1,2], **Yun Young Choi**[*,1], **Joohwan Ko**[3], **Minseon Gwak**[†,4]

[1] SolverX, [2] Seoul National University, [3] University of Massachusetts Amherst,
[4] Pohang University of Science and Technology

## Abstract

Modeling long spatiotemporal sequences requires capturing both complex spatial correlations and temporal dependencies. Convolutional State Space Models (ConvSSMs) have been proposed to incorporate spatial modeling in State Space Models (SSMs) using the convolution of tensor-valued states and kernels. Yet, existing implementations remain limited to $1 \times 1$ state kernels for computational feasibility, which limits the modeling capacity of ConvSSMs. We introduce a novel spatiotemporal model, ConvT3 (**Conv**SSM using **T**ridiagonal **T**oeplitz **T**ensors), designed to equivalently realize ConvSSMs with extended $3 \times 3$ state kernels. ConvT3 structures a state kernel for its corresponding tensor to be composed as a structured SSM matrix on hidden state dimensions and a constrained tridiagonal Toeplitz tensor on spatial dimensions. We show that the structured tensor can be diagonalized, which enables efficient parallel training while leveraging $3 \times 3$ state convolutions. We demonstrate that ConvT3 effectively embeds rich spatial and temporal information into the dynamics of tensor-valued states, achieving state-of-the-art performance on most metrics in long-range video generation and physical system modeling.

## 1 Introduction

Modeling spatiotemporal dynamics is foundational for numerous domains, including videos (Finn et al., 2016; Ho et al., 2022; Wasim et al., 2024), physical systems (Nguyen et al., 2023; Li et al., 2024; Choi et al., 2020), and weather forecasting (Lam et al., 2023; Bodnar et al., 2024; Pathak et al., 2022). In video generation, maintaining long-term memory and consistency is crucial for ensuring realistic outputs. Similarly, modeling physical systems such as fluid flow or thermodynamic processes requires fine-grained spatial reasoning within temporal dynamics, as spatial interactions evolve in such systems. Consequently, spatiotemporal modeling has drawn intense interest as a key challenge for domains that require capturing intricate dependencies across space and time.

Across visual and physical domains, sequence models offer a compelling paradigm for spatiotemporal modeling, capturing the underlying dynamics that link input sequences to output sequences. The primary objective of spatiotemporal sequence modeling is to integrate spatial feature extraction with temporal representation. Early approaches like Convolutional Recurrent Neural Networks (ConvRNNs), exemplified by ConvLSTM (Shi et al., 2015), introduced tensor-valued hidden states, updated with convolutions, to capture spatial patterns. However, ConvRNNs inherit limitations of RNNs, including serial training and difficulty in modeling long-range dependencies. Transformer-based models (Yan et al., 2023; Park et al., 2023; Pătrăucean et al., 2024) have also been adapted to video domains, employing factorized attention or patched-base processing to handle spatiotemporal inputs. Although powerful, the quadratic-time computational cost of attention restricts the feasible spatiotemporal context, especially as the data dimensionality and spatial/temporal resolution increase.

Convolutional State Space Models (ConvSSMs) (Smith et al., 2023) combine tensor-valued states from ConvRNN with State Space Models (SSMs), providing expressive modeling capability with linear-time efficiency. Although ConvSSMs conceptually allow arbitrary kernel sizes for the state,

---

[*]Equal contribution
[†]Corresponding author: minseon25@postech.ac.kr

input, output, and feedthrough convolutions, ConvS5 (Smith et al., 2023), the practical implementation of ConvSSM, restricts the state kernels to pointwise $1 \times 1$ convolutions. This constraint is necessary to avoid exploding computation in parallel scans with larger kernels, but it fundamentally limits the learned state dynamics from effectively capturing spatiotemporal context. Such limitations motivate alternative designs that can leverage extended state kernels while maintaining efficient training.

In this paper, we propose a ConvSSM using Tridiagonal Toeplitz Tensors (ConvT3), which equivalently implements a ConvSSM with $3 \times 3$ state convolution. We first reformulate the convolution operations in a ConvSSM as tensor contractions, where state, input, output, and feedthrough tensors correspond to their respective kernels. The key idea of ConvT3 is to structure the state tensor using two components: a diagonalizable SSM matrix and a constrained tridiagonal Toeplitz tensor. The proposed structuring rule ensures ConvT3's diagonalizability and correspondence to the extended $3 \times 3$ kernels. Finally, we develop an efficient training algorithm using linear-complexity parallel scans with stable parameterization, which naturally generalizes to higher-dimensional data.

We empirically validate the effectiveness of ConvT3 on modeling videos and physical dynamics. On the long-range Moving-MNIST benchmark (Srivastava et al., 2015), ConvT3 consistently outperforms existing sequence models and achieves state-of-the-art results in video generation. On PDEBench datasets (Takamoto et al., 2022), ConvT3 attains the best accuracy in physical system modeling, while also exhibiting superior training stability compared to ConvS5. Ablation studies further confirm that the modeling performance of ConvT3 arises from its structured $3 \times 3$ state kernels rather than parameter growth. Together, these results demonstrate that ConvT3 provides a scalable, stable, and effective framework for modeling complex spatiotemporal dynamics.

## 2 PRELIMINARIES

Before introducing the equations, we summarize in Table 7 the symbols used for values and operations, with notations distinguished according to the type of value.

### 2.1 CONVOLUTIONAL STATE SPACE MODELS WITH POINTWISE STATE KERNELS

**ConvSSM** A ConvSSM using general-size convolution kernels was conceptually suggested by Smith et al. (2023), where an SSM combines convolutional operations for spatial feature extraction. Let $\mathcal{U}(t) \in \mathbb{R}^{H \times W \times U}$ be a $U$-channel two-dimensional data at time $t$, where $H$ and $W$ denote spatial height and width. A continuous-time ConvSSM is formulated with a state tensor $\mathcal{X}(t) \in \mathbb{C}^{H \times W \times P}$, where $P$ denotes the hidden state dimension, and an output tensor $\mathcal{Y}(t) \in \mathbb{R}^{H \times W \times U}$, as

$$\mathcal{X}'(t) = \boldsymbol{\mathcal{A}} * \mathcal{X}(t) + \boldsymbol{\mathcal{B}} * \mathcal{U}(t), \tag{1}$$

$$\mathcal{Y}(t) = \boldsymbol{\mathcal{C}} * \mathcal{X}(t) + \boldsymbol{\mathcal{D}} * \mathcal{U}(t), \tag{2}$$

where $*$ denotes zero-padded convolution that preserves the tensor shapes, and $\mathcal{X}'(t) \triangleq \frac{d}{dt}\mathcal{X}(t)$. The convolution kernels $\boldsymbol{\mathcal{A}} \in \mathbb{C}^{P \times P \times k_A \times k_A}$, $\boldsymbol{\mathcal{B}} \in \mathbb{C}^{P \times U \times k_B \times k_B}$, $\boldsymbol{\mathcal{C}} \in \mathbb{C}^{U \times P \times k_C \times k_C}$, $\boldsymbol{\mathcal{D}} \in \mathbb{C}^{U \times U \times k_D \times k_D}$ are referred to as the state, input, output, and feedthrough kernels, respectively. The input–output tensor sequence can remain real-valued despite operating in the complex domain by parameterizing the kernels with conjugate pairs Gu et al. (2022).

**ConvS5** A ConvS5 (Smith et al., 2023) implements a ConvSSM with a *pointwise state kernel*, i.e., $k_A = 1$, to apply parallel scans to the discretized ConvSSM

$$\mathcal{X}_{k+1} = \overline{\boldsymbol{\mathcal{A}}} * \mathcal{X}_k + \overline{\boldsymbol{\mathcal{B}}} * \mathcal{U}_k, \tag{3}$$

$$\mathcal{Y}_k = \boldsymbol{\mathcal{C}} * \mathcal{X}_k + \boldsymbol{\mathcal{D}} * \mathcal{U}_k, \tag{4}$$

where $\overline{\boldsymbol{\mathcal{A}}} \in \mathbb{C}^{P \times P \times 1 \times 1}$ and $\overline{\boldsymbol{\mathcal{B}}} \in \mathbb{C}^{P \times U \times k_B \times k_B}$ represent the state and input kernels discretized for a timescale parameter $\Delta \in \mathbb{R}^P$ by discretization methods, such as zero-order hold. The computation of the state tensor sequence $\mathcal{X}_{1:L}$ from the input tensor sequence $\mathcal{U}_{1:L}$ is enabled by parallel scans for an element $q_k = (q_{k,a}, q_{k,b}) := (\overline{\boldsymbol{\mathcal{A}}}, \overline{\boldsymbol{\mathcal{B}}} * \mathcal{U}_k)$ and a binary associative operator $\bullet$ defined by

$$q_i \bullet q_j := (q_{j,a} \circ q_{i,a}, \; q_{j,a} * q_{i,b} + q_{j,b}), \tag{5}$$

where $\circ$ denotes the convolution of kernels and $+$ is elementwise addition. Since the operation $\circ$ produces growing kernels across scans, ConvS5 parameterizes $\boldsymbol{\mathcal{A}} \in \mathbb{C}^{P \times P \times 1 \times 1}$ to maintain training feasibility on long sequences by preventing kernel growth during parallel scans.

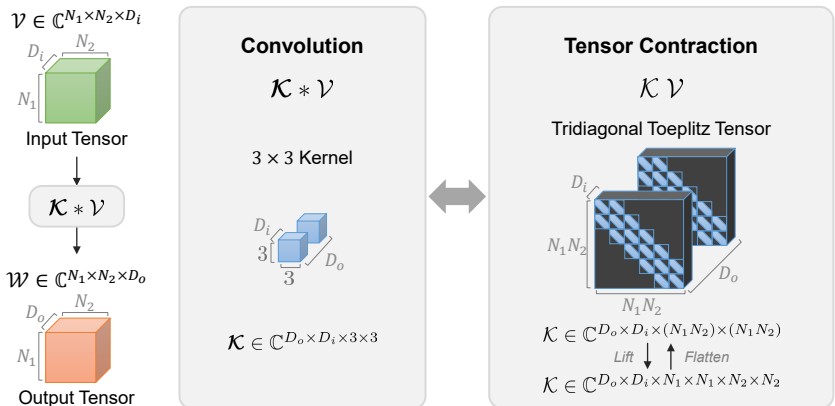

Figure 1: Equivalence between $3 \times 3$ convolution and its tensor contraction formulation using a TT tensor. For interpretability, the TT tensor is depicted with block TT matrices in the flattened form. The convolution operator $\mathcal{K} * \mathcal{V}$ can be expressed as a structured tensor contraction $\mathcal{K}\mathcal{V}$.

However, this design restricts the expressivity of the state dynamics by reducing $\mathcal{A}$ to a pointwise state kernel, leaving the spatial modeling to $\mathcal{B}, \mathcal{C}, \mathcal{D}$ kernels and to deeper stacking of layers.

## 2.2 Eigendecomposition of Tridiagonal Toeplitz Matrices

As a preliminary, we recall the eigendecomposition of **tridiagonal Toeplitz (TT)** matrices. Define $T = \text{tridiag}(l_T, d_T, u_T) \in \mathbb{C}^{N \times N}$, where $l_T, d_T, u_T \in \mathbb{C}$ are the lower, diagonal, and upper entries:

$$T = \text{tridiag}(l_T, d_T, u_T) = \begin{bmatrix} d_T & u_T & & & \\ l_T & d_T & u_T & & \\ & \ddots & \ddots & \ddots & \\ & & l_T & d_T & u_T \\ & & & l_T & d_T \end{bmatrix}. \tag{6}$$

It is well known that a TT matrix admits a *closed-form eigendecomposition*, with the $i$th eigenvalue $\lambda_i$ and its corresponding eigenvector $x_i$ are given by

$$\lambda_i = d_T + 2\sqrt{l_T u_T} \cos\left(\tfrac{i\pi}{N+1}\right), \tag{7}$$

$$x_{ij} = (l_T/u_T)^{j/2} \sin\left(\tfrac{ij\pi}{N+1}\right). \tag{8}$$

where $x_{ij}$ denotes the $j$th entry of the $i$th eigenvector, for $i, j = \{1, \ldots, N\}$.

Thus, one can obtain the eigenvalues and eigenvectors of $T$ using its entries $l_T, d_T, u_T$ without further computation. In particular, TT matrices with the same off-diagonal ratio share a common eigenbasis.

## 2.3 Convolution and Tridiagonal Toeplitz Tensors

Convolutions are linear and shift-invariant, so they can be rewritten as matrix/tensor operations with structured matrices/tensors. In 1D, they correspond to multiplication with Toeplitz matrices; in higher dimensions, to tensor contraction with Toeplitz tensors. The Toeplitz structure arises from the shift-invariance, while the convolution kernel size determines the number of nonzero off-diagonals.

For a $3 \times 3$ kernel $\mathcal{K} \in \mathbb{C}^{D_o \times D_i \times 3 \times 3}$ (with $D_o$ output and $D_i$ input channels), 2D convolutions can be written as the tensor contractions with TT tensors. Specifically, we call $\mathcal{T} \in \mathbb{C}^{D_o \times D_i \times N_1 \times N_1 \times N_2 \times N_2}$ a **TT tensor** if each slice $\mathcal{T}_{q,r,i_1,j_1,:,:} \in \mathbb{C}^{N_2 \times N_2}$ and $\mathcal{T}_{q,r,:,:,i_2,j_2} \in \mathbb{C}^{N_1 \times N_1}$ is TT matrix for all $q \in \{1, \ldots, D_o\}$, $r \in \{1, \ldots, D_i\}$, $i_1, j_1 \in \{1, \ldots, N_1\}$, $i_2, j_2 \in \{1, \ldots, N_2\}$ such that $|i_1 - j_1| \le 1$, $|i_2 - j_2| \le 1$. Values outside the tridiagonal patterns are zero.

Then, for an input tensor $\mathcal{V} \in \mathbb{C}^{N_1 \times N_2 \times D_i}$, the convolution with $\mathcal{K}$ can be equivalently written as

$$\mathcal{K} * \mathcal{V} = \mathcal{K}\mathcal{V} \in \mathbb{C}^{N_1 \times N_2 \times D_o}, \tag{9}$$

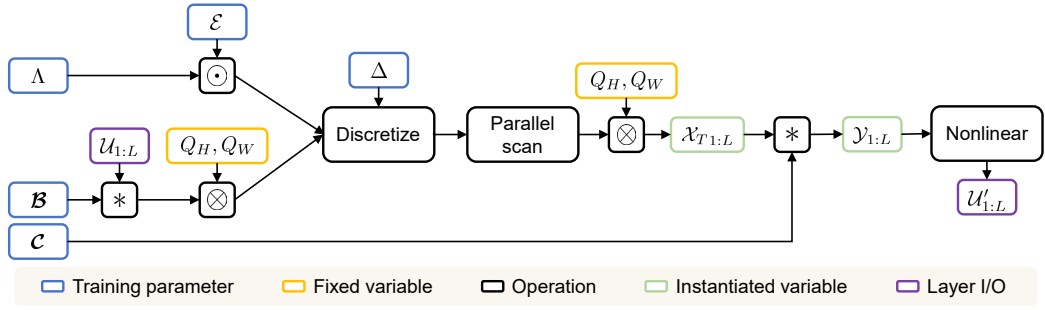

Figure 2: Algorithmic flow of parallel computation in a ConvT3 layer for length-$L$ input and output tensor sequences $\mathcal{U}_{1:L}$, $\mathcal{U}'_{1:L} \in \mathbb{R}^{L \times H \times W \times U}$. The proportionality constraints in ConvT3 induce a form that allows parallel scans to be applied.

where $\mathcal{K} \in \mathbb{C}^{D_o \times D_i \times N_1 \times N_1 \times N_2 \times N_2}$ is the associated TT tensor whose entries are induced by $\mathcal{K}$. The operation between the two tensors is a matrix-multiplication-like contraction along each $D_i$, $N_1$, $N_2$ dimension. This correspondence between convolution and tensor contraction is illustrated in Figure 1.

## 3 METHOD

In this section, three types of tensor operations are employed: (1) Elementwise multiplication along all dimensions, denoted by $\odot$; (2) Tensor product resulting in dimension extension, denoted by $\otimes$; and (3) Tensor contraction over certain dimensions, where the operator symbol is omitted. The contraction dimensions are specified using Einstein notation in Appendix A.1.

### 3.1 STRUCTURED STATE KERNELS OF CONVT3

Our goal is to construct a ConvSSM state kernel $\mathcal{A}$ of size larger than $1 \times 1$, while having a diagonalizable form, thus enabling linear complexity parallel scan. Also, we wish to employ the structure of a well-known state matrix like S5 along the state dimension to guarantee performance. We operate with a diagonalizable state matrix and a structured TT tensor to achieve such goals. We define the state tensor $\mathcal{A}$ corresponding to the state kernel $\mathcal{A}$ by the following method:

$$\mathcal{A} := f(R, \mathcal{S}), \tag{10}$$

where $R \in \mathbb{C}^{P \times P}$ is a diagonalizable matrix, $\mathcal{S} \in \mathbb{C}^{P \times P \times H \times H \times W \times W}$ is a **proportionality-constrained TT (PTT)** tensor, and $f$ denotes the composition rule to construct a state tensor. Specifically, the PTT tensor $\mathcal{S}$ satisfies two proportionality conditions with some nonzero ratios $\alpha_H, \alpha_W \in \mathbb{C}$:

$$\text{(i)} \quad l_{\mathcal{S}_{q,r,:,:,i_w,j_w}} = \alpha_H \, u_{\mathcal{S}_{q,r,:,:,i_w,j_w}}, \qquad \text{(ii)} \quad l_{\mathcal{S}_{q,r,i_h,j_h,:,:}} = \alpha_W \, u_{\mathcal{S}_{q,r,i_h,j_h,:,:}}, \tag{11}$$

for all $q, r \in \{1, \ldots, P\}$, $i_h, j_h \in \{1, \ldots, H\}$, $i_w, j_w \in \{1, \ldots, W\}$ such that $|i_h - j_h| \leq 1$, $|i_w - j_w| \leq 1$.

Moreover, for efficient computation in (14), we impose that the PTT tensor is diagonal along the hidden $P \times P$ dimension, i.e., each slice $\mathcal{S}_{:,:,i_h,j_h,i_w,j_w}$ is diagonal.

Importantly, we can decompose $R$ and $S$ using their properties. First, since $R$ is diagonalizable, there exists an invertible matrix $Q_P \in \mathbb{C}^{P \times P}$ such that

$$R = Q_P \Lambda Q_P^{-1}, \tag{12}$$

where $\Lambda \in \mathbb{C}^{P \times P}$ is diagonal.

Next, since (8) and (11), the spatial slices share common eigenbases $Q_H \in \mathbb{C}^{H \times H}$ and $Q_W \in \mathbb{C}^{W \times W}$ for the height and width dimensions, where the eigenbases are uniquely determined by $\alpha_H$ and $\alpha_W$. Thus, $\mathcal{S}$ can be decomposed as

$$\mathcal{S} = (I_P \otimes Q_H \otimes Q_W) \, \mathcal{E} \, (I_P \otimes Q_H \otimes Q_W)^{-1} \in \mathbb{C}^{P \times P \times H \times H \times W \times W}, \tag{13}$$

where $I_P \in \mathbb{R}^{P \times P}$ denotes the identity matrix, $\mathcal{E} \in \mathbb{C}^{P \times P \times H \times H \times W \times W}$, such that each slice $\mathcal{E}_{:,:,i_h,j_h,i_w,j_w} \in \mathbb{C}^{P \times P}$, $\mathcal{E}_{q,r,:,:,i_w,j_w} \in \mathbb{C}^{H \times H}$ and $\mathcal{E}_{q,r,i_h,j_h,:,:} \in \mathbb{C}^{W \times W}$ are diagonal for $q, r \in \{1, \ldots, P\}$, $i_h, j_h \in \{1, \ldots, H\}$, $i_w, j_w \in \{1, \ldots, W\}$ (Refer to Appendix A.2 for derivation).

Based on the decomposed factors, the composition rule $f$ is defined as follows:

$$f\big(R_{(Q_P,\Lambda)}, S_{(Q_H,Q_W,\mathcal{E})}\big) = (Q_P \otimes Q_H \otimes Q_W)\big((\Lambda \otimes I_H \otimes I_W) \odot \mathcal{E}\big)(Q_P \otimes Q_H \otimes Q_W)^{-1}. \quad (14)$$

We formally define the proposed model, ConvT3, as Definition 1.

**Definition 1.** A ConvT3 is defined as

$$\begin{aligned} \mathcal{X}'(t) &= \mathcal{A}\,\mathcal{X}(t) + \mathcal{B}\,\mathcal{U}(t), \\ \mathcal{Y}(t) &= \mathcal{C}\,\mathcal{X}(t) + \mathcal{D}\,\mathcal{U}(t), \end{aligned} \quad (15)$$

where $\mathcal{A} = \mathcal{Q}\big((\Lambda \otimes I_H \otimes I_W) \odot \mathcal{E}\big)\mathcal{Q}^{-1} \in \mathbb{C}^{P \times P \times H \times H \times W \times W}$ constructed by the composition rule $f$ in (14), $\mathcal{B} \in \mathbb{C}^{P \times U \times H \times H \times W \times W}$, $\mathcal{C} \in \mathbb{C}^{U \times P \times H \times H \times W \times W}$, and $\mathcal{D} \in \mathbb{C}^{U \times U \times H \times H \times W \times W}$.

We next show that the ConvT3 defined above is equivalent to a ConvSSM with a $3 \times 3$ state kernel.

**Theorem 1.** A ConvT3 is a ConvSSM with a $3 \times 3$ state kernel $\mathbf{\mathcal{A}} \in \mathbb{C}^{P \times P \times 3 \times 3}$.

The key idea behind the proof (Appendix A.3) is that the state tensor of ConvT3, defined as

$$\mathcal{A} = (Q_P \otimes Q_H \otimes Q_W)\big((\Lambda \otimes I_H \otimes I_W) \odot \mathcal{E}\big)(Q_P \otimes Q_H \otimes Q_W)^{-1}, \quad (16)$$

retains a PTT structure. The middle factor $(\Lambda \otimes I_H \otimes I_W) \odot \mathcal{E}$, contracted with $Q_H$ and $Q_W$, is itself a PTT tensor, since $\Lambda$ is block-diagonal and $\mathcal{E}$ preserves the Proportional Toeplitz pattern along the spatial dimensions $H$ and $W$. The contraction with $Q_P$ does not break this structure: $Q_P$ also only acts on channel dimensions. As a result, the full tensor $\mathcal{A}$ is a PTT tensor. By the equivalence between 2D convolution operations and tensor contractions with TT tensors (9), this establishes that ConvT3 is indeed a ConvSSM with a $3 \times 3$ state kernel.

## 3.2 PARALLEL TRAINING OF CONVT3 IN LINEAR TIME

We next show that ConvT3 can be diagonalized, enabling the use of linear-time parallel scan in sequence length (Proof is provided in Appendix A.4).

**Theorem 2.** A ConvT3 can be diagonalized as

$$\begin{aligned} \mathcal{X}'_T(t) &= \mathcal{A}_T \mathcal{X}_T(t) + \mathcal{B}_T\,\mathcal{U}(t), \\ \mathcal{Y}(t) &= \mathcal{C}_T \mathcal{X}_T(t) + \mathcal{D}\,\mathcal{U}(t), \end{aligned} \quad (17)$$

where

$$\mathcal{A}_T = (\Lambda \otimes I_H \otimes I_W) \odot \mathcal{E}, \qquad \mathcal{B}_T = \mathcal{Q}^{-1}B, \qquad \mathcal{C}_T = C\mathcal{Q}, \quad (18)$$

under the change of state $\mathcal{X}_T(t) = \mathcal{Q}^{-1}\mathcal{X}(t)$, with $\mathcal{Q} := Q_P \otimes Q_H \otimes Q_W$. The contraction dimensions of tensor contraction are stated by Einstein notation in Appendix A.1.

Since the transformed state tensor $\mathcal{A}_T$ (18) is diagonal, parallel scans using the operator in (5) can be applied to the transformed ConvT3 (17), enabling linear-time complexity in sequence length. See Appendix B for full complexity analysis.

For equivalent state dynamics, transformation on $\mathcal{B}$ and $\mathcal{C}$ using $\mathcal{Q}$ are performed before and after the scan. In practice, the transformation $Q_P$ for the hidden dimension is omitted, since the state along the $P$ dimension can be assumed to be trained in a diagonalized form, analogous to diagonal SSMs. This implementation detail is essential for reducing computational complexity, as changing the kernel operation of $\mathcal{B}, \mathcal{C}$ to tensor products would be inefficient. Thus, the effective transformation reduces to the spatial transformation $Q_H \otimes Q_W$, which is sufficient for preserving the desired equivalence. The overall algorithmic flow for parallel training of ConvT3 is shown in Figure 2.

## 3.3 PARAMETERIZATION OF CONVT3 FOR TRAINING STABILITY

**Overview of Learnable Parameters and Stable Reparameterization.** In continuous SSMs, stability follows from the Hurwitz condition applied to the diagonalized state matrix, since negative real parts

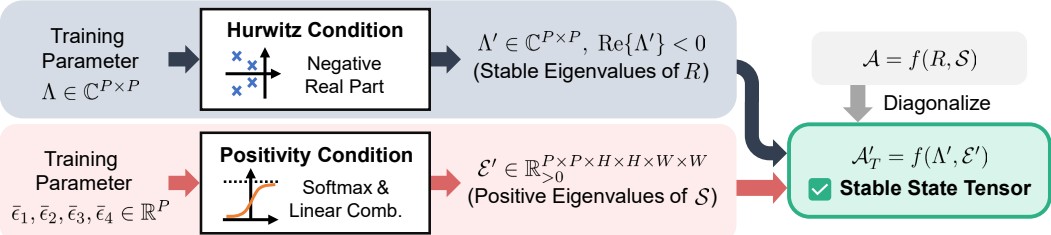

Figure 3: Reparameterization of ConvT3 state kernel for stability. The $\Lambda$ (eigenvalues of $R$) is conditioned to have negative real parts, while the $\mathcal{E}$ (eigenvalues of $\mathcal{S}$) is conditioned to be positive. This ensures the stability of the resulting state tensor constructed from their combinations. Here, $\bar{\epsilon}_i$ denotes the vector whose components are the $\epsilon_i$ from all effective channels, and $'$ indicates that the value is reparameterized.

guarantee contractive temporal dynamics. For ConvT3, the same principle applies, i.e., the stability of $\mathcal{A}_T$ is ensured when the real part of its diagonal values remains strictly negative.

As shown in Figure 3, the diagonalized state tensor $\mathcal{A}_T$ can attain stable dynamics by enforcing two conditions: (1) $\Lambda \in \mathbb{C}^{P \times P}$ has negative real parts, and (2) $\mathcal{E}$ are strictly positive. These conditions ensure the negativity of $\text{Re}\{\mathcal{A}_T\}$, and thus guarantee the stability of ConvT3. We now describe the reparameterization methods used to ensure that these conditions are satisfied.

**Hurwitz Condition.** To guarantee the negativity of real part of $\Lambda$ during training, we reparameterize the learnable $\Lambda$ as

$$\Lambda' \; : \; \text{Re}\{\Lambda'\} = -\text{softplus}(\text{Re}\{\Lambda\}),$$

which automatically keeps $\Re\{\Lambda'\} < 0$ at all times.

**Positivity Condition.** The other condition for stability is strict positivity of $\mathcal{E} > 0$, where $\mathcal{E}$ is the eigenvalue of the PTT tensor $\mathcal{S} \in \mathbb{C}^{P \times P \times H \times H \times W \times W}$.

We construct $\mathcal{E}$ by extending the Toeplitz eigenvalue formula (7) to the PTT tensor by

$$\mathcal{E}_{q,r,i_h,j_h,i_w,j_w} = \epsilon_{q,r}(\theta_{i_h}^H, \theta_{i_w}^W)\, \delta_{q,r} \delta_{i_h,j_h} \delta_{i_w,j_w}, \tag{19}$$

where

$$\epsilon_{q,r}(\theta_{i_h}^H, \theta_{i_w}^W) = a_{q,r} + b_{q,r} \cos\theta_{i_h}^H + c_{q,r} \cos\theta_{i_w}^W + d_{q,r} \cos\theta_{i_h}^H \cos\theta_{i_w}^W, \tag{20}$$

for $\theta_{i_h}^H = \frac{\pi i_h}{H+1}$ and $\theta_{i_w}^W = \frac{\pi i_w}{W+1}$, with the channel indices $q, r \in \{1, \ldots, P\}$, the spatial indices $i_h, j_h \in \{1, \ldots, H\}$ and $i_w, j_w \in \{1, \ldots, W\}$, and the Kronecker delta $\delta_{\cdot,\cdot}$. The scalar coefficients $a_{q,r}, b_{q,r}, c_{q,r}, d_{q,r} \in \mathbb{C}$ along with the off-diagonal proportions induces the $3 \times 3$ state kernel, where $a_{q,r}$ corresponds to the *center value* and $b_{q,r}, c_{q,r}, d_{q,r}$ with the *side values* of the kernel.

For simplicity, we drop the indices $(q, r)$ with the understanding that the following derivations apply to all $(q, r)$. Thus, (20) is rewritten as

$$\epsilon(\theta_{i_h}^H, \theta_{i_w}^W) := a + b \cos\theta_{i_h}^H + c \cos\theta_{i_w}^W + d \cos\theta_{i_h}^H \cos\theta_{i_w}^W. \tag{21}$$

To guarantee positivity for all discrete $(\theta_{i_h}^H, \theta_{i_w}^W)$, it suffices to impose positivity over the continuous domain $(0, \pi) \times (0, \pi)$. Since $\epsilon$ is bilinear in $\cos\theta^H, \cos\theta^W \in (-1, 1)$, positivity is enforced by checking the four extreme points:

$$\epsilon_1 = a + b + c + d > 0, \;\; \epsilon_2 = a + b - c - d > 0,$$

$$\epsilon_3 = a - b + c - d > 0, \;\; \epsilon_4 = a - b - c + d > 0.$$

Here, we can fix the center coefficient $a$ to 1, reducing redundant degrees of freedom, which implies

$$\epsilon_1 + \epsilon_2 + \epsilon_3 + \epsilon_4 = 4.$$

To satisfy all positivity constraints automatically, we reparameterize the four extreme values via a softmax:

$$\epsilon_i' = 4 \cdot \text{softmax}(\epsilon_1, \epsilon_2, \epsilon_3, \epsilon_4)_i,$$

which ensures $\epsilon_i' > 0$ and $\sum_i \epsilon_i' = 4$. The coefficients $(b, c, d)$ are then recovered using the inverse linear transformation:

$$b = (\epsilon_1' + \epsilon_2' - \epsilon_3' - \epsilon_4')/4, \quad c = (\epsilon_1' - \epsilon_2' + \epsilon_3' - \epsilon_4')/4, \quad d = (\epsilon_1' - \epsilon_2' - \epsilon_3' + \epsilon_4')/4.$$

**Stable State Tensor $\mathcal{A}_T$.** Finally, combining the Hurwitz-stable $\Lambda'$ and the positive eigenvalues $\mathcal{E}'$ yields a diagonalized state operator that satisfies $\text{Re}\{\mathcal{A}_T\} < 0$ elementwise. Thus, the ConvT3 state tensor is guaranteed to remain stable throughout training while retaining fully learnable temporal and spatial dynamics.

### 3.4 GENERALIZATION TO $N$-DIMENSIONAL CONVT3

The correspondence between convolution and Toeplitz tensors can extend ConvT3 to higher dimensions. While 2D convolutions are represented by TT tensors for two spatial dimensions, $N$-dimensional convolutions naturally induce TT tensors for $N$ spatial dimensions. In a higher-order TT tensor, the tridiagonal pattern also arises along each of the $N$ spatial axes for an $N$-dimensional kernel with a kernel size of 3. Thus, for an input tensor in an $N$-dimensional space, a convolution can be represented as a tensor contraction with the $N$-dimensional TT tensor followed by the contraction over the channel and spatial indices.

This implies that the ConvT3 formulation, defined for 2D inputs and PTT tensors, admits a natural theoretical extension to arbitrary spatial dimensions. The proportionality-constrained structure and parallel scan mechanism remain applicable, as they depend only on the PTT property along each axis rather than the dimensionality itself.

## 4 RELATED WORKS

**ConvRNN-based Models** ConvRNNs capture local spatiotemporal correlations and are widely used in video prediction and physical simulations (Ballas et al., 2015); early variants such as ConvLSTM (Shi et al., 2015) and PredRNN (Wang et al., 2017) improved prediction by integrating spatial–temporal cues and introducing spatiotemporal memory. Building on these, SwinLSTM (Tang et al., 2023) embeds Transformer blocks within LSTM cells to model local and global dependencies, and a later PredRNN (Wang et al., 2022) improves long-term stability via memory decoupling.

**Transformer-based Models** Transformers, with self-attention, excel at global dependency modeling and are widely used in spatiotemporal tasks (Lee et al., 2024; Vaswani et al., 2017). Recent variants include TECO, which preserves long-term temporal consistency via a VQ-based non-autoregressive framework (Yan et al., 2023); PredFormer, a purely Transformer model with competitive results across benchmarks (Tang et al., 2024b); and TRecViT, a hybrid that pairs lightweight temporal recurrence with spatial Transformer blocks to balance efficiency and accuracy on long sequences (Pătrăucean et al., 2024).

**SSM-based Models** SSMs provide linear-time sequence modeling with strong memory retention (Zhang et al., 2024; Huang et al., 2025; Liu et al., 2024). Early S4 (Gu et al., 2021) and S5 (Smith et al., 2022) paved the way; ConvSSM (Smith et al., 2023) added spatial modeling via convolutions; S4ND (Nguyen et al., 2022) extended SSMs to multidimensional signals for vision; Selective SSM (Wang et al., 2023) prunes information-sparse tokens; and VMRNN (Tang et al., 2024a) leverages Mamba modules (Gu & Dao, 2023) to capture short- and long-term dynamics.

## 5 EXPERIMENTS

In Section 5.1, we evaluate the spatiotemporal modeling capability of ConvT3 on a long-range video generation task. In Section 5.2, we evaluate ConvT3 on complex physical system modeling. In Section 5.3, we present ablation studies on architecture and hyperparameters.

For all tasks, we used off-diagonal proportions $\alpha_H = \alpha_W = -1$, meaning the state kernel is skew-symmetric across spatial dimensions. We initialized $b_{q,r}, c_{q,r}, d_{q,r}$ to zero to ensure equivalence with

Table 1: Evaluation on the Moving-MNIST dataset (Srivastava et al., 2015). We condition on 100 frames, and then show results after generating 800 and 1200 frames. Bold: best performance. Underline: second-best performance.

**Trained on 300 frames**

| Method | FVD ↓ | $100 \rightarrow 800$ PSNR ↑ | SSIM ↑ | LPIPS ↓ | FVD ↓ | $100 \rightarrow 1200$ PSNR ↑ | SSIM ↑ | LPIPS ↓ |
|---|---|---|---|---|---|---|---|---|
| Transformer (Vaswani et al., 2017) | 159 | 12.6 | 0.609 | 0.287 | 265 | 12.4 | 0.591 | 0.321 |
| Performer (Choromanski et al., 2021) | 234 | 13.4 | 0.652 | 0.379 | 275 | 13.2 | 0.592 | 0.393 |
| CW-VAE (Saxena et al., 2021) | 104 | 12.4 | 0.592 | 0.277 | **117** | 12.3 | 0.585 | 0.286 |
| ConvLSTM (Shi et al., 2015) | 128 | 15.0 | 0.737 | 0.169 | 187 | 14.1 | 0.706 | 0.203 |
| ConvS5 (Smith et al., 2023) | **72** | 16.0 | 0.761 | 0.156 | 187 | 14.5 | 0.678 | 0.230 |
| ConvT3 | 79 | **16.1** | **0.776** | **0.146** | 118 | **15.2** | **0.746** | **0.179** |

**Trained on 600 frames**

| Method | FVD ↓ | PSNR ↑ | SSIM ↑ | LPIPS ↓ | FVD ↓ | PSNR ↑ | SSIM ↑ | LPIPS ↓ |
|---|---|---|---|---|---|---|---|---|
| Transformer | 42 | 13.7 | 0.672 | 0.207 | 91 | 13.1 | 0.631 | 0.252 |
| Performer | 93 | 12.4 | 0.616 | 0.274 | 243 | 12.2 | 0.608 | 0.312 |
| CW-VAE | 94 | 12.5 | 0.598 | 0.269 | 107 | 12.3 | 0.590 | 0.280 |
| ConvLSTM | 91 | 15.5 | 0.757 | 0.149 | 137 | 14.6 | 0.727 | 0.180 |
| ConvS5 | 47 | 16.4 | 0.788 | 0.134 | 71 | 15.6 | 0.763 | 0.162 |
| ConvT3 | **36** | **17.7** | **0.823** | **0.104** | **56** | **16.7** | **0.795** | **0.131** |

ConvS5 at initialization. The other experiment setups for each task are provided in Appendix D. Implementation is available at `https://github.com/voltwin-dev/ConvT3`.

## 5.1 LONG-RANGE VIDEO MODELING AND GENERATION

Video modeling requires capturing both spatial structures within each frame and temporal dynamics across long sequences. To evaluate such capability, we consider the Moving-MNIST dataset (Srivastava et al., 2015), a standard benchmark where digits move within a 2D frame.

Following the task setup in ConvS5 (Smith et al., 2023), we train models to reconstruct 300 or 600 input frames, and evaluate them at test time by 400, 800, and 1200 future frames from 100 observed frames. Baselines include Transformer (Vaswani et al., 2017), Performer (Choromanski et al., 2021), ConvLSTM (Shi et al., 2015), CW-VAE (Saxena et al., 2021), and ConvS5.

ConvT3 consistently outperforms existing baselines, showing significant improvements across nearly all metrics. When trained on 300 frames, ConvT3 achieves the best PSNR, SSIM, and LPIPS scores and second-best FVD scores for both 800- and 1200-frame generation. With longer training sequences of 600 frames, ConvT3 further amplifies its advantage, achieving the best scores for all metrics and prediction lengths. Overall, ConvT3 demonstrates state-of-the-art performance on the Moving-MNIST benchmark, validating the effectiveness of structured $3 \times 3$ state kernels for long-range spatiotemporal modeling.

## 5.2 PHYSICAL SYSTEM MODELING

### 5.2.1 PARTIAL DIFFERENTIAL EQUATION MODELING

Physical system modeling requires capturing the underlying spatiotemporal dynamics that govern complex phenomena, often expressed as PDEs. To assess ConvT3 on accurate prediction of physical dynamics, we use the PDEBench dataset (Takamoto et al., 2022), following the prediction task setup in MPP (McCabe et al., 2024). Among the 2D datasets in PDEBench, we exclude the computationally heavy compressible Navier–Stokes case and focus on the *Shallow-Water* and *Diffusion-Reaction* datasets, which require accurate modeling of nonlinear PDE dynamics.

For training, models are provided with the first 16 time steps of grid trajectories and optimized to predict the next single step. For ConvS5 and ConvT3 in this task, we replaced the attention layers in AViT by ConvS5 and ConvT3 layers, allowing direct comparison of spatiotemporal modeling ability within the same backbone. Performance is measured in terms of normalized root mean square error (NRMSE), along with inference time. Baselines include FNO (Li et al., 2020), UNet (Ronneberger et al., 2015), and AViT (McCabe et al., 2024), and ConvS5.

Table 2: Evaluation on Shallow-Water and Diffusion-Reaction datasets (Takamoto et al., 2022). Complex parameters were counted as two reals. Inference time was measured on A100 GPU. Bold: best performance. Underline: second-best performance. Dash: Not provided by source.

| Model | #Params | NRMSE ↓ | | Time (s) | |
|---|---|---|---|---|---|
| | | Shallow-Water | Diffusion-Reaction | Train Step | Evaluation Step |
| AViT-B | 116M | 0.00047 | 0.0110 | - | - |
| FNO-B | 115M | 0.00246 | 0.0599 | - | - |
| UNet | 7M | 0.083− | 0.84− | - | - |
| FNO | 927K | 0.0044 | 0.12− | - | - |
| AViT-Ti | 7M | 0.00053 | 0.0090 | 303 (2.31×) | 2.74 (2.06×) |
| ConvS5 | 6M | 0.00035 | 0.0106 | 131 (1.00×) | 1.33 (1.00×) |
| ConvT3 | 6M | **0.00033** | **0.0087** | 151 (1.15×) | 1.51 (1.14×) |

As shown in Table 2, ConvT3 achieved the best accuracy on both Shallow-Water and Diffusion-Reaction datasets while using significantly fewer parameters compared to large baselines. On Shallow-Water, ConvT3 performs comparably to ConvS5, while on Diffusion-Reaction, ConvT3 achieved a substantial performance gain over ConvS5. Moreover, ConvT3 maintains efficiency close to ConvS5, demonstrating both effectiveness and scalability in modeling complex physical dynamics.

### 5.2.2 TRAINING STABILITY

ConvS5 often exhibited training instability, whereas ConvT3 remained stable under the same experimental and model configurations. As one representative instance shown in Figure 4, ConvT3 maintains a smooth loss curve, while ConvS5 suddenly spikes. This behavior persisted across multiple random seeds, suggesting it is not from initialization effects.

### 5.3 MODEL ABLATIONS

We conduct an ablation study on the standard Moving-MNIST task (Srivastava et al., 2015), predicting 10 frames from 10 observed frames.

In Table 3, we evaluate a variant of ConvT3, named MiniT3, which shares the same kernel slice across $P \times P$ hidden dimensions. This variant introduces structural alternation from ConvS5 to ConvT3, while minimizing the additional learnable parameter. Specifically, ConvT3 increases $3P$ parameters, but MiniT3 only increases 3 parameters per layer. Notably, MiniT3 significantly outperformed ConvS5 with only 24 additional parameters, demonstrating that the superior performance of ConvT3 arises from its structural improvements rather than parameter increase.

In Table 4, we check the influence of the off-diagonal proportions $\alpha_H$ and $\alpha_W$ with skew-symmetric and symmetric state kernels. The results showed nearly identical performance, implying that the proportions can be set to fixed, arbitrary values.

In Table 5 and Table 6, we compare ConvS5 and ConvT3 on various numbers of ConvSSM blocks and $\mathcal{B}, \mathcal{C}$ kernel sizes. ConvT3 outperformed ConvS5 in most cases, especially when either $\mathcal{B}$ or $\mathcal{C}$ kernels were $1 \times 1$ sized, and thus the spatial modeling effect was minimal.

## 6 DISCUSSION

This work proposed ConvT3, which extends the state kernels of ConvSSM from pointwise to $3 \times 3$. This extension enriched the spatial representation within the state dynamics and enabled the model to capture more complex interactions. As a result, ConvT3 achieved significant performance gains in downstream tasks. Another strength of ConvT3 lies in its efficient training algorithm. The algorithm scales linearly with sequence length, which makes ConvT3 feasible for long-range modeling. Experimental evaluations confirmed the inference speed, highlighting that the proposed method maintains efficiency without compromising modeling power. Furthermore, the parameters were carefully designed with stability in mind. By constraining the parameter space, ConvT3 avoided

Table 3: (**Ablations: Minimal parameterization.**) MiniT3 outperformed ConvS5 despite the minimal increase in parameter numbers.

| Model | #Params | MSE ↓ | MAE ↓ |
|---|---|---|---|
| ConvS5 | 24M | 11.57 | 23.25 |
| MiniT3 | 24M (**+24**) | **10.87** | **21.64** |

Table 4: (**Ablations: Off-diagonal proportion.**) Different proportions gave similar results.

| $\alpha_H, \alpha_W$ | MSE ↓ | MAE ↓ |
|---|---|---|
| 1 (Symmetric) | 10.97 | 22.12 |
| −1 (Skew-symmetric) | 10.99 | 22.15 |

Table 5: (**Ablations: Number of blocks.**) ConvT3 generally achieved better performance than ConvS5 under the same number of blocks.

| Model | #Blocks | MSE ↓ | MAE ↓ |
|---|---|---|---|
| ConvS5 | 2 | 13.15 | 27.08 |
| | 4 | 11.92 | **24.64** |
| | 6 | 11.13 | **22.76** |
| | 8 | 11.57 | 23.25 |
| | 12 | 10.68 | 21.80 |
| ConvT3 | 2 | **12.91** | **25.75** |
| | 4 | **11.82** | 25.31 |
| | 6 | **11.11** | 23.38 |
| | 8 | **10.99** | **22.15** |
| | 12 | **10.57** | **21.77** |

Table 6: (**Ablations: Kernel size.**) The introduction of a $3 \times 3$ state kernel $\mathcal{A}$, in contrast to relying solely on same-sized $\mathcal{B}$ and $\mathcal{C}$ kernels, resulted in notable performance gains. Bold: better performance under the same $\mathcal{B}$ and $\mathcal{C}$ settings. (More cases are provided in Table 12.)

| Model | $\mathcal{A}$ kernel | $\mathcal{B}$ kernel | $\mathcal{C}$ kernel | MSE ↓ | MAE ↓ |
|---|---|---|---|---|---|
| ConvS5 | $1 \times 1$ | $1 \times 1$ | $1 \times 1$ | 14.98 | 30.60 |
| | $1 \times 1$ | $1 \times 1$ | $3 \times 3$ | 12.74 | 24.07 |
| | $1 \times 1$ | $3 \times 3$ | $1 \times 1$ | 12.13 | 24.35 |
| | $1 \times 1$ | $3 \times 3$ | $3 \times 3$ | 11.57 | 23.25 |
| ConvT3 | $3 \times 3$ | $1 \times 1$ | $1 \times 1$ | **14.68** | **29.65** |
| | $3 \times 3$ | $1 \times 1$ | $3 \times 3$ | **11.17** | **23.04** |
| | $3 \times 3$ | $3 \times 3$ | $1 \times 1$ | **11.69** | **23.46** |
| | $3 \times 3$ | $3 \times 3$ | $3 \times 3$ | **10.99** | **22.15** |

unstable dynamics that can hinder learning. This stability-oriented design was reflected in the experiments, where the model consistently exhibited smooth convergence and across datasets.

In theory, ConvT3 can be generalized to $N$-dimensional data by extending the PTT construction as described in Section 3.4. However, our empirical validation was conducted on 2D datasets, which are the most prevalent and practically important modalities. The absence of experiments on higher-dimensional data remains a limitation of this study. Furthermore, we adopted fixed proportionality conditions for the parameterization of PTT tensors. While this choice ensured tractability, it may limit flexibility. A natural extension would be to allow these proportionalities to be learnable parameters, potentially improving adaptability while retaining structural interpretability.

ETHICS STATEMENT

This research was conducted in compliance with ethical guidelines. At the current stage of the study, no foreseeable ethical risks or societal harms have been identified.

REPRODUCIBILITY STATEMENT

We provide a link to access the source code. For theoretical results, all assumptions are clearly stated, and complete proofs are included in the appendix.

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

# Supplementary Material

## A  PROOFS

| Notation | Definition |
|---|---|
| $a$ | Scalar |
| $A$ | Matrix |
| $\mathbf{A}$ | Block matrix |
| $\mathcal{A}$ | Tensor |
| $\boldsymbol{\mathcal{A}}$ | Convolution kernel |
| $*$ | Convolution (Kernel–Tensor) |
| $\circ$ | Convolution (Kernel–Kernel) |
| $\odot$ | Elementwise multiplication |
| $\otimes$ | Tensor product |
| $\oplus$ | Direct sum |
| $\delta_{.,.}$ | Kronecker delta |

Table 7: Major notations

### A.1  EXPLICIT FORM OF TENSOR CONTRACTIONS IN SECTION 3

The explicit tensor contractions are described by Einstein notation as follows:

- Equation (12)

$$R_{q,r} = Q^P_{q,s} \Lambda_{s,t} (Q^P)^{-1}_{t,r}.$$

- Equation (13)

$$\mathcal{S}_{q,r,i_h,j_h,i_w,j_w} = Q^H_{i_h,k_h} Q^W_{i_w,k_w} \mathcal{E}_{q,r,k_h,l_h,k_w,l_w} (Q^H)^{-1}_{l_h,j_h} (Q^W)^{-1}_{l_w,j_w}.$$

- Equation (16)

$$\mathcal{A}_{q,r,i_h,j_h,i_w,j_w} = Q^P_{q,s} Q^H_{i_h,k_h} Q^W_{i_w,k_w} (\Lambda_{s,t} \odot \mathcal{E}_{s,t,k_h,l_h,k_w,l_w}) (Q^P)^{-1}_{t,r} (Q^H)^{-1}_{l_h,j_h} (Q^W)^{-1}_{l_w,j_w}.$$

- Equation (18), Diagonalized state tensor

$$\mathcal{A_T}_{q,r,i_h,j_h,i_w,j_w} = \Lambda_{q,r} \odot \mathcal{E}_{q,r,i_h,j_h,i_w,j_w}.$$

- Equation (18), Transformation tensor

$$\mathcal{Q}_{q,r,i_h,j_h,i_w,j_w} = Q^P_{q,r} Q^H_{i_h,j_h} Q^W_{i_w,j_w}.$$

### A.2  PROOF OF (13)

We derive (13) using the following theorems for general TT and PTT tensors. In the proofs of the theorems, we use block matrices, i.e., flattened forms of tensors, for clarity of exposition.

We first introduce the definition of a commutation matrix, which will be used in Theorem 3 to show that a block TT matrix admits a closed-form eigendecomposition when its outer blocks share a common eigenvector.

**Definition 2.** The $(m,n)$-commutation matrix $K^{(m,n)} \in \{0,1\}^{mn \times mn}$ is the unique permutation matrix satisfying $K^{(m,n)}\mathrm{vec}(A) = \mathrm{vec}(A^T)$ for all $A \in \mathbb{C}^{m \times n}$.

**Theorem 3.** Let $\mathbf{T} \in \mathbb{C}^{nm \times nm}$ be a block TT matrix with blocks $L, D, U \in \mathbb{C}^{m \times m}$ sharing a common eigenbasis. Then $\mathbf{T}$ admits a *closed-form eigendecomposition* of the form

$$\mathbf{T} = \widetilde{\mathbf{Q}} \, \overline{\boldsymbol{\Lambda}}' \, \widetilde{\mathbf{Q}}^{-1}, \tag{22}$$

where $\widetilde{\mathbf{Q}}$ and $\overline{\boldsymbol{\Lambda}}'$ are explicitly constructed in the proof.

*Proof of Theorem 3.* Suppose that the blocks admit eigendecompositions of the form

$$L = Q\Lambda_L Q^{-1}, \quad D = Q\Lambda_D Q^{-1}, \quad U = Q\Lambda_U Q^{-1},$$

with a common eigenvector matrix $Q$ and diagonal matrices $\Lambda_L = \text{diag}(\lambda_{L,1}, \cdots, \lambda_{L,m})$, $\Lambda_D = \text{diag}(\lambda_{D,1}, \cdots, \lambda_{D,m})$, and $\Lambda_U = \text{diag}(\lambda_{U,1}, \cdots, \lambda_{U,m})$. Then, $\mathbf{T}$ can be written with a block tridiagonal matrix $\mathbf{\Lambda}$ as

$$\mathbf{T} = \mathbf{Q}\mathbf{\Lambda}\mathbf{Q}^{-1}, \tag{23}$$

where $\mathbf{Q} = \text{diag}(Q, \ldots, Q)$ and $\mathbf{\Lambda} = \text{tridiag}(\Lambda_L, \Lambda_D, \Lambda_U)$.

Now, define a permuted matrix $\mathbf{\Lambda}'$ by

$$\mathbf{\Lambda}' := K^{(m,n)}\mathbf{\Lambda}K^{(n,m)}, \tag{24}$$

where $K^{(m,n)} \in \{0,1\}^{mn \times mn}$ is the $(m,n)$-commutation matrix. The permuted matrix $\mathbf{\Lambda}'$ decomposes as the direct sum of tridiagonal Toeplitz matrices $T_i$ for $i = 1, \cdots, m$,

$$\mathbf{\Lambda}' = T_1 \oplus \cdots \oplus T_m,$$

where $\oplus$ denotes the direct sum and each component $T_i \in \mathbb{C}^{n \times n}$ is given by $T_i = \text{tridiag}(\lambda_{L,i}, \lambda_{D,i}, \lambda_{U,i})$.

Since $T_i$ is a TT matrix, it has a closed-form eigendecomposition,

$$T_i = \overline{Q}_i\overline{\Lambda}_i\overline{Q}_i^{-1},$$

where $\overline{\Lambda}_i = \text{diag}(\mu_{i,1} \cdots \mu_{i,n})$ is a diagonal eigenvalue matrix with $\mu_{i,k} = \lambda_{D,i} + 2\sqrt{\lambda_{L,i}\lambda_{U,i}}\cos(\frac{k\pi}{n+1})$ and $\overline{Q}_i = [v_1^i, \cdots, v_n^i]$ is the matrix of the corresponding eigenvectors with $v_k^i(j) = (\lambda_{L,i}/\lambda_{U,i})^{k/2}\sin\frac{jk\pi}{n+1}$.

Since the permuted matrix $\mathbf{\Lambda}'$ is the direct sum of $T_i$ for $i = 1, \cdots, m$, $\mathbf{\Lambda}'$ has a closed-form eigendecomposition of the form

$$\mathbf{\Lambda}' = \overline{\mathbf{Q}}\,\overline{\mathbf{\Lambda}}\,\overline{\mathbf{Q}}^{-1}, \tag{25}$$

where $\overline{\mathbf{Q}} = \overline{Q}_1 \oplus \cdots \oplus \overline{Q}_m$ and $\overline{\mathbf{\Lambda}} = \overline{\Lambda}_1 \oplus \cdots \oplus \overline{\Lambda}_m$.

Combining (23)–(25), the block TT matrix $\mathbf{T}$ has a closed-form eigendecomposition since $K^{(m,n)}K^{(n,m)} = I_{mn}$.

$$\mathbf{T} = \widetilde{\mathbf{Q}}\,\overline{\mathbf{\Lambda}}'\,\widetilde{\mathbf{Q}}^{-1}, \tag{26}$$

where $\widetilde{\mathbf{Q}} := \mathbf{Q}\,\overline{\mathbf{Q}}'$, $\overline{\mathbf{Q}}' := K^{(n,m)}\overline{\mathbf{Q}}K^{(m,n)}$, and $\overline{\mathbf{\Lambda}}' := K^{(n,m)}\overline{\mathbf{\Lambda}}K^{(m,n)}$. $\qquad\square$

Now, we obtain Theorem 4 when constraining the proportionality of the block entries.

**Theorem 4.** Let $\mathbf{T} \in \mathbb{C}^{nm \times nm}$ be a block TT matrix with blocks $L, D, U \in \mathbb{C}^{m \times m}$ sharing a common eigenbasis $Q$. If $L = \alpha U$ for arbitrary $\alpha \in \mathbb{C}$, then $\mathbf{T}$ admits a closed-form eigendecomposition of the form

$$\mathbf{T} = \begin{bmatrix} \overline{Q}_{11}Q & \cdots & \overline{Q}_{1n}Q \\ \vdots & \ddots & \vdots \\ \overline{Q}_{n1}Q & \cdots & \overline{Q}_{nn}Q \end{bmatrix}\overline{\mathbf{\Lambda}}'\begin{bmatrix} \overline{Q}_{11}Q & \cdots & \overline{Q}_{1n}Q \\ \vdots & \ddots & \vdots \\ \overline{Q}_{n1}Q & \cdots & \overline{Q}_{nn}Q \end{bmatrix}^{-1}, \tag{27}$$

where $\overline{Q}$ depends only on the matrix size $n$ and $\alpha$.

*Proof.* Using the notation of Theorem 3, when $L = \alpha U$, each TT matrix $T_i = \text{tridiag}(\lambda_{L,i}, \lambda_{D,i}, \lambda_{U,i})$ satisfies $\lambda_{L,i} = \alpha\lambda_{U,i}$. By (8), this implies that all $T_i$ share the same eigenbasis $\overline{Q}$, i.e., $\overline{Q} = \overline{Q}_1 = \cdots = \overline{Q}_m \in \mathbb{C}^{m \times m}$. Therefore, $\overline{\mathbf{Q}} = \text{diag}(\overline{Q}, \ldots, \overline{Q}) \in \mathbb{C}^{nm \times nm}$ is uniquely determined by $\alpha$. Since $\mathbf{Q} = \text{diag}(Q, \ldots, Q) \in \mathbb{C}^{nm \times nm}$, substituting into the decomposition (26) yields (27), proving the claim. $\qquad\square$

Since the PTT tensor $\mathcal{S}$ in (13) has proportionality conditions for both spatial dimensions, applying Theorem 4 to $\mathcal{S}$ yields Corollary 4.1, which represents the explicit form of tensor contractions in (13).

**Corollary 4.1.** A PTT tensor $\mathcal{S} \in \mathbb{C}^{P \times P \times H \times H \times W \times W}$ can be decomposed as

$$\mathcal{S}_{q,r,i_1,j_1,i_2,j_2} = Q^H_{i_1,k_1}Q^W_{i_2,k_2}\,\mathcal{E}^P_{q,r,k_1,l_1,k_2,l_2}\,(Q^H)^{-1}_{l_1,j_1}(Q^W)^{-1}_{l_2,j_2},$$

where $Q^H \in \mathbb{C}^{H \times H}$, $Q^W \in \mathbb{C}^{W \times W}$, and $\mathcal{E}^P \in \mathbb{C}^{P \times P \times H \times H \times W \times W}$.

### A.3 PROOF OF THEOREM 1

*Proof.* We first show that the state tensor $\mathcal{A}$ defined in (1) is a TT tensor.

$$
\begin{aligned}
\mathcal{A}_{q,r,i_h,j_h,i_w,j_w} &= Q_{q,s}^P Q_{i_h,k_h}^H Q_{i_w,k_w}^W (\Lambda_{s,t} \odot \mathcal{E}_{s,t,k_h,l_h,k_w,l_w})(Q^P)_{t,r}^{-1}(Q^H)_{l_h,j_h}^{-1}(Q^W)_{l_w,j_w}^{-1} \\
&= Q_{q,s}^P(\Lambda_{s,t} \odot \mathcal{S}_{s,t,i_h,j_h,i_w,j_w})(Q^P)_{t,r}^{-1}
\end{aligned}
\tag{28}
$$

For all $i_h, j_h \in \{1, \ldots, H\}$ such that $|i_h - j_h| > 1$, all slices $\mathcal{A}_{:,:,i_h,j_h,:,:}$ are zero, since $\mathcal{S}$ has zero value, which gives the tridiagonality condition for the $H$ dimension. Likewise, the state tensor $\mathcal{A}$ is tridiagonal along the $W$ dimension.

For all $i_h, i_h' \in \{1, \ldots, H-1\}$, the following equation holds due to the Toeplitz condition for $\mathcal{S}$.

$$
\begin{aligned}
\mathcal{A}_{q,r,i_h,i_h+1,i_w,j_w} &= Q_{q,s}^P(\Lambda_{s,t} \odot \mathcal{S}_{s,t,i_h,i_h+1,i_w,j_w})(Q^P)_{t,r}^{-1}, \\
&= Q_{q,s}^P(\Lambda_{s,t} \odot \mathcal{S}_{s,t,i_h',i_h'+1,i_w,j_w})(Q^P)_{t,r}^{-1}, \\
&= \mathcal{A}_{q,r,i_h',i_h'+1,i_w,j_w}, \\
&= Q_{q,s}^P.
\end{aligned}
\tag{29}
$$

Likewise, the condition is satisfied for the lower diagonal and the $W$ dimension, proving the Toeplitz condition for $\mathcal{A}$.

Since (9) and $\mathcal{A}$ is a TT tensor, it is equivalent to a $P \times P \times 3 \times 3$ kernel operation. □

### A.4 PROOF OF THEOREM 2

*Proof.* By definition (1), ConvT3 is given by

$$
\begin{aligned}
\mathcal{X}'(t) &= \mathcal{Q}\big((\Lambda \otimes I_H \otimes I_W) \odot \mathcal{E}\big)\mathcal{Q}^{-1}\mathcal{X}(t) + \mathcal{B}\,\mathcal{U}(t), \\
\mathcal{Y}(t) &= \mathcal{C}\,\mathcal{X}(t) + \mathcal{D}\,\mathcal{U}(t).
\end{aligned}
\tag{30}
$$

Suppose a state transformation $\mathcal{X}_T(t) = \mathcal{Q}^{-1}\mathcal{X}(t)$. Substituting $\mathcal{X}(t) = \mathcal{Q}\mathcal{X}_T(t)$ yields

$$
\begin{aligned}
\mathcal{Q}\mathcal{X}_T'(t) &= \mathcal{Q}\big((\Lambda \otimes I_H \otimes I_W) \odot \mathcal{E}\big)\mathcal{Q}^{-1}\mathcal{Q}\mathcal{X}_T(t) + \mathcal{B}\mathcal{U}(t), \\
\mathcal{Y}(t) &= \mathcal{C}\,\mathcal{Q}\mathcal{X}_T(t) + \mathcal{D}\mathcal{U}(t).
\end{aligned}
\tag{31}
$$

By performing tensor contraction with $\mathcal{Q}^{-1}$, we obtain

$$
\mathcal{X}_T'(t) = \big((\Lambda \otimes I_H \otimes I_W) \odot \mathcal{E}\big)\mathcal{X}_T(t) + \mathcal{Q}^{-1}\mathcal{B}\mathcal{U}(t).
\tag{32}
$$

Therefore, the transformed system is

$$
(\mathcal{A}_T, \mathcal{B}_T, \mathcal{C}_T, \mathcal{D}) = \big((\Lambda \otimes I_H \otimes I_W) \odot \mathcal{E}, \ \mathcal{Q}^{-1}\mathcal{B}, \ \mathcal{C}\mathcal{Q}, \ \mathcal{D}\big),
$$

which completes the proof.

□

## B  COMPUTATIONAL COMPLEXITY OF CONVT3

The computational complexity of parallel scan in ConvT3, given input $\mathcal{B}_T \, \mathcal{U}_{1:L} \in \mathbb{C}^{L \times H \times W \times P}$ is $\mathcal{O}(LPHW)$, identical to the ConvSSM model with pointwise state kernel. This is because the diagonalized form of the state tensor $\mathcal{A}$ allows identical parallel scan operations for the two cases.

The total computational complexity for a single pointwise state kernel ConvSSM layer is $\mathcal{O}(LPHW + T_B + T_C)$, with $T_B$ the cost for input kernel convolution, and $T_C$ the cost for output kernel convolution. Since $T_B = \mathcal{O}(LPUk_B^2 HW)$ and $T_c = \mathcal{O}(LPUk_C^2 HW)$, the total cost will be $\mathcal{O}(LPHWU(k_B^2 + k_c^2))$ (Smith et al., 2023).

For ConvT3, we have additional matrix multiplication operations with $Q_H$ and $Q_W$ (Check Figure 2 for details), so the total cost is $\mathcal{O}(LPHWU(k_B^2 + k_C^2) + LPHW(H + W))$ if naively implemented. However, matrix multiplication with the $Q$ matrix is the same as a discrete sine transform (DST) type-I, and thus the matrix multiplication can be implemented via fast Fourier transform (FFT)-based routines. Instead of $\mathcal{O}(H^2)$ complexity due to dense matrix multiplication, the DST type-I has a complexity of $\mathcal{O}(H \log H)$. Thus, with FFT-based routines, the total complexity of ConvT3 is reduced to $\mathcal{O}(LPHWU(k_B^2 + k_c^2) + LPHW(\log H + \log W))$.

Notice two main things: First, the additional computational cost of ConvT3 does not scale with the input dimension $U$, which is normally the same scale as the state dimension $P$, meaning ConvT3 becomes more efficient as the hidden dimension of the model grows. Second, although the cost of ConvT3 scales more according to spatial size compared to ConvS5, the cost ratio grows logarithmically, thus there does not exist a point where ConvT3 becomes unfeasible.

Table 8, Table 9, and Table 10 provide inference-time and memory comparisons between single ConvS5 and ConvT3 layers across different state sizes, hidden dimensions, sequence lengths, and image resolutions, respectively.

Table 8: Inference time and memory usage under different state sizes.

| State size $P$, Hidden dim $U$ | Time [ms] | | Memory [GB] | |
|---|---|---|---|---|
| | ConvS5 | ConvT3 | ConvS5 | ConvT3 |
| 512 | 52.7 (1.00×) | 65.8 (1.25×) | 4.26 (1.00×) | 5.22 (1.23×) |
| 256 | 20.1 (1.00×) | 28.0 (1.39×) | 2.13 (1.00×) | 2.62 (1.23×) |
| 128 | 10.5 (1.00×) | 13.8 (1.31×) | 1.09 (1.00×) | 1.34 (1.23×) |
| 64 | 7.2 (1.00×) | 9.3 (1.29×) | 0.58 (1.00×) | 0.70 (1.21×) |

Table 9: Inference time and memory usage under different sequence lengths.

| Sequence length $L$ | Time [ms] | | Memory [GB] | |
|---|---|---|---|---|
| | ConvS5 | ConvT3 | ConvS5 | ConvT3 |
| 100 | 100.2 (1.00×) | 138.6 (1.38×) | 10.46 (1.00×) | 12.39 (1.18×) |
| 50 | 50.8 (1.00×) | 67.3 (1.32×) | 5.25 (1.00×) | 6.28 (1.19×) |
| 20 | 20.9 (1.00×) | 27.9 (1.33×) | 2.13 (1.00×) | 2.62 (1.22×) |
| 10 | 11.9 (1.00×) | 15.4 (1.29×) | 1.09 (1.00×) | 1.40 (1.28×) |

Table 10: Inference time and memory usage under different image resolutions.

| Image resolution $H \times W$ | Time [ms] | | Memory [GB] | |
|---|---|---|---|---|
| | ConvS5 | ConvT3 | ConvS5 | ConvT3 |
| 32×32 | 77.2 (1.00×) | 160.6 (2.08×) | 8.25 (1.00×) | 11.75 (1.42×) |
| 16×16 | 20.9 (1.00×) | 27.8 (1.33×) | 2.13 (1.00×) | 2.62 (1.23×) |
| 8×8 | 7.7 (1.00×) | 9.7 (1.26×) | 0.60 (1.00×) | 0.70 (1.16×) |
| 4×4 | 7.0 (1.00×) | 8.0 (1.14×) | 0.22 (1.00×) | 0.24 (1.08×) |

# C SUPPLEMENTARY RESULTS

Table 11: Full results on the Moving-MNIST dataset (Srivastava et al., 2015). The number of training frames, 300 or 600, is specified in the table header. The evaluation task is to condition on 100 frames, and then generate forward 400, 800, and 1200 frames.

**Trained on 300 frames**

| Method | Params | FVD ↓ | $100 \rightarrow 400$ PSNR ↑ | SSIM ↑ | LPIPS ↓ |
|---|---|---|---|---|---|
| Transformer | 164M | $73 \pm 3$ | $13.5 \pm 0.1$ | $0.669 \pm 0.002$ | $0.213 \pm 0.003$ |
| Performer | 164M | $111 \pm 9$ | $13.4 \pm 0.1$ | $0.653 \pm 0.002$ | $0.288 \pm 0.001$ |
| CW-VAE | 20M | $78 \pm 1$ | $12.7 \pm 0.1$ | $0.611 \pm 0.002$ | $0.254 \pm 0.001$ |
| ConvLSTM | 20M | $57 \pm 3$ | $16.9 \pm 0.2$ | $0.796 \pm 0.004$ | $0.113 \pm 0.002$ |
| ConvSSM (random init) | 20M | $67 \pm 3$ | $15.5 \pm 0.1$ | $0.742 \pm 0.001$ | $0.168 \pm 0.001$ |
| ConvS5 | 20M | $26 \pm 1$ | $18.1 \pm 0.1$ | $0.830 \pm 0.003$ | $0.094 \pm 0.002$ |
| ConvS5 (reproduced) | 21M | $26 \pm 2$ | $17.9 \pm 0.1$ | $0.824 \pm 0.003$ | $0.097 \pm 0.001$ |
| ConvT3 | 21M | $33 \pm 2$ | $18.0 \pm 0.1$ | $0.828 \pm 0.003$ | $0.096 \pm 0.002$ |

**Trained on 600 frames**

| Method | Params | FVD ↓ | $100 \rightarrow 400$ PSNR ↑ | SSIM ↑ | LPIPS ↓ |
|---|---|---|---|---|---|
| Transformer | 164M | $21 \pm 1$ | $15.0 \pm 0.1$ | $0.741 \pm 0.002$ | $0.138 \pm 0.001$ |
| Performer | 164M | $27 \pm 1$ | $13.1 \pm 0.1$ | $0.654 \pm 0.004$ | $0.206 \pm 0.001$ |
| CW-VAE | 20M | $73 \pm 2$ | $12.9 \pm 0.1$ | $0.621 \pm 0.004$ | $0.242 \pm 0.001$ |
| ConvLSTM | 20M | $39 \pm 5$ | $17.3 \pm 0.2$ | $0.812 \pm 0.005$ | $0.100 \pm 0.003$ |
| ConvSSM (random init) | 20M | $81 \pm 6$ | $15.5 \pm 0.1$ | $0.743 \pm 0.002$ | $0.163 \pm 0.003$ |
| ConvS5 | 20M | $23 \pm 3$ | $18.1 \pm 0.1$ | $0.832 \pm 0.003$ | $0.092 \pm 0.003$ |
| ConvS5 (reproduced) | 21M | $15 \pm 1$ | $19.5 \pm 0.2$ | $0.865 \pm 0.004$ | $0.071 \pm 0.003$ |
| ConvT3 | 21M | $16 \pm 1$ | $19.8 \pm 0.1$ | $0.871 \pm 0.002$ | $0.066 \pm 0.000$ |

**Trained on 300 frames**

| Method | Params | FVD ↓ | $100 \rightarrow 800$ PSNR ↑ | SSIM ↑ | LPIPS ↓ |
|---|---|---|---|---|---|
| Transformer | 164M | $159 \pm 7$ | $12.6 \pm 0.1$ | $0.609 \pm 0.002$ | $0.287 \pm 0.001$ |
| Performer | 164M | $234 \pm 1$ | $13.4 \pm 0.1$ | $0.652 \pm 0.006$ | $0.379 \pm 0.002$ |
| CW-VAE | 20M | $104 \pm 2$ | $12.4 \pm 0.1$ | $0.592 \pm 0.002$ | $0.277 \pm 0.002$ |
| ConvLSTM | 20M | $128 \pm 4$ | $15.0 \pm 0.1$ | $0.737 \pm 0.003$ | $0.169 \pm 0.001$ |
| ConvSSM (random init) | 20M | $287 \pm 5$ | $13.6 \pm 0.1$ | $0.577 \pm 0.001$ | $0.293 \pm 0.001$ |
| ConvS5 | 20M | $72 \pm 3$ | $16.0 \pm 0.1$ | $0.761 \pm 0.005$ | $0.156 \pm 0.003$ |
| ConvS5 (reproduced) | 21M | $74 \pm 3$ | $16.0 \pm 0.1$ | $0.767 \pm 0.004$ | $0.152 \pm 0.001$ |
| ConvT3 ) | 21M | $79 \pm 2$ | $16.1 \pm 0.1$ | $0.776 \pm 0.004$ | $0.146 \pm 0.002$ |

**Trained on 600 frames**

| Method | Params | FVD ↓ | $100 \rightarrow 800$ PSNR ↑ | SSIM ↑ | LPIPS ↓ |
|---|---|---|---|---|---|
| Transformer | 164M | $42 \pm 2$ | $13.7 \pm 0.1$ | $0.672 \pm 0.002$ | $0.207 \pm 0.003$ |
| Performer | 164M | $93 \pm 5$ | $12.4 \pm 0.1$ | $0.616 \pm 0.002$ | $0.274 \pm 0.001$ |
| CW-VAE | 20M | $94 \pm 3$ | $12.5 \pm 0.9$ | $0.598 \pm 0.004$ | $0.269 \pm 0.001$ |
| ConvLSTM | 20M | $91 \pm 7$ | $15.5 \pm 0.2$ | $0.757 \pm 0.004$ | $0.149 \pm 0.003$ |
| ConvSSM (random init) | 20M | $145 \pm 8$ | $14.3 \pm 0.1$ | $0.696 \pm 0.002$ | $0.218 \pm 0.002$ |
| ConvS5 | 20M | $23 \pm 3$ | $18.1 \pm 0.1$ | $0.832 \pm 0.003$ | $0.092 \pm 0.003$ |
| ConvS5 (reproduced) | 21M | $35 \pm 3$ | $17.6 \pm 0.2$ | $0.819 \pm 0.005$ | $0.109 \pm 0.005$ |
| ConvT3 | 21M | $36 \pm 4$ | $17.7 \pm 0.1$ | $0.823 \pm 0.003$ | $0.104 \pm 0.001$ |

**Trained on 300 frames**

| Method | Params | FVD ↓ | $100 \rightarrow 1200$ PSNR ↑ | SSIM ↑ | LPIPS ↓ |
|---|---|---|---|---|---|
| Transformer | 164M | $265 \pm 8$ | $12.4 \pm 0.1$ | $0.591 \pm 0.002$ | $0.321 \pm 0.002$ |
| Performer | 164M | $275 \pm 5$ | $13.2 \pm 0.1$ | $0.592 \pm 0.001$ | $0.393 \pm 0.001$ |
| CW-VAE | 20M | $117 \pm 2$ | $12.3 \pm 0.1$ | $0.585 \pm 0.002$ | $0.286 \pm 0.001$ |
| ConvLSTM | 20M | $187 \pm 6$ | $14.1 \pm 0.1$ | $0.706 \pm 0.003$ | $0.203 \pm 0.001$ |
| ConvSSM (random init) | 20M | $511 \pm 8$ | $13.3 \pm 0.1$ | $0.515 \pm 0.001$ | $0.348 \pm 0.001$ |
| ConvS5 | 20M | $187 \pm 5$ | $14.5 \pm 0.1$ | $0.678 \pm 0.003$ | $0.230 \pm 0.004$ |
| ConvS5 (reproduced) | 21M | $130 \pm 5$ | $14.9 \pm 0.1$ | $0.721 \pm 0.004$ | $0.198 \pm 0.002$ |
| ConvT3 | 21M | $118 \pm 3$ | $15.2 \pm 0.1$ | $0.746 \pm 0.004$ | $0.179 \pm 0.002$ |

**Trained on 600 frames**

| Method | Params | FVD ↓ | $100 \rightarrow 1200$ PSNR ↑ | SSIM ↑ | LPIPS ↓ |
|---|---|---|---|---|---|
| Transformer | 164M | $91 \pm 6$ | $13.1 \pm 0.1$ | $0.631 \pm 0.004$ | $0.252 \pm 0.002$ |
| Performer | 164M | $243 \pm 7$ | $12.2 \pm 0.1$ | $0.608 \pm 0.001$ | $0.312 \pm 0.002$ |
| CW-VAE | 20M | $107 \pm 2$ | $12.3 \pm 0.1$ | $0.590 \pm 0.004$ | $0.280 \pm 0.002$ |
| ConvLSTM | 20M | $137 \pm 9$ | $14.6 \pm 0.1$ | $0.727 \pm 0.004$ | $0.180 \pm 0.003$ |
| ConvSSM (random init) | 20M | $215 \pm 9$ | $13.4 \pm 0.1$ | $0.614 \pm 0.001$ | $0.287 \pm 0.001$ |
| ConvS5 | 20M | $71 \pm 9$ | $15.6 \pm 0.1$ | $0.763 \pm 0.002$ | $0.162 \pm 0.003$ |
| ConvS5 (reproduced) | 21M | $55 \pm 4$ | $16.6 \pm 0.1$ | $0.791 \pm 0.005$ | $0.136 \pm 0.005$ |
| ConvT3 | 21M | $56 \pm 7$ | $16.7 \pm 0.1$ | $0.795 \pm 0.003$ | $0.131 \pm 0.002$ |

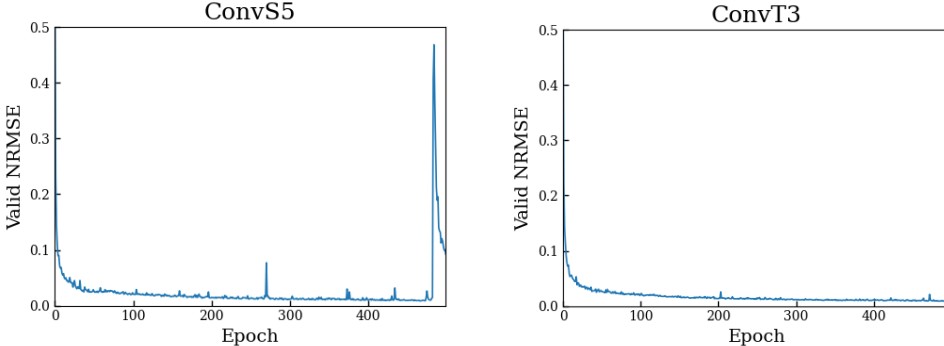

Figure 4: Training loss curves of ConvT3 and ConvS5 in a representative run on the Diffusion Reaction dataset, illustrating their training stability.

Table 12: (**Ablations: Kernel size.**) The introduction of a $3 \times 3$ state kernel $\mathcal{A}$, in contrast to relying solely on same-sized $\mathcal{B}$ and $\mathcal{C}$ kernels, resulted in notable performance gains. Bold: better performance under the same $\mathcal{B}$ and $\mathcal{C}$ settings.

| Model | $\mathcal{A}$ kernel | $\mathcal{B}$ kernel | $\mathcal{C}$ kernel | MSE ↓ | MAE ↓ |
|---|---|---|---|---|---|
| ConvS5 | $1 \times 1$ | $1 \times 1$ | $1 \times 1$ | 14.98 | 30.60 |
|  | $1 \times 1$ | $1 \times 1$ | $3 \times 3$ | 12.74 | 24.07 |
|  | $1 \times 1$ | $1 \times 1$ | $5 \times 5$ | 10.56 | 21.03 |
|  | $1 \times 1$ | $3 \times 3$ | $1 \times 1$ | 12.13 | 24.35 |
|  | $1 \times 1$ | $3 \times 3$ | $3 \times 3$ | 11.57 | 23.25 |
|  | $1 \times 1$ | $3 \times 3$ | $5 \times 5$ | 11.49 | 22.83 |
|  | $1 \times 1$ | $5 \times 5$ | $1 \times 1$ | 11.82 | 23.52 |
|  | $1 \times 1$ | $5 \times 5$ | $3 \times 3$ | **12.48** | **24.43** |
|  | $1 \times 1$ | $5 \times 5$ | $5 \times 5$ | 13.56 | 24.73 |
| ConvT3 | $3 \times 3$ | $1 \times 1$ | $1 \times 1$ | **14.68** | **29.65** |
|  | $3 \times 3$ | $1 \times 1$ | $3 \times 3$ | **11.17** | **23.04** |
|  | $3 \times 3$ | $1 \times 1$ | $5 \times 5$ | **9.91** | **20.38** |
|  | $3 \times 3$ | $3 \times 3$ | $1 \times 1$ | **11.69** | **23.46** |
|  | $3 \times 3$ | $3 \times 3$ | $3 \times 3$ | **10.99** | **22.15** |
|  | $3 \times 3$ | $3 \times 3$ | $5 \times 5$ | **11.34** | **22.43** |
|  | $3 \times 3$ | $5 \times 5$ | $1 \times 1$ | **11.48** | **23.09** |
|  | $3 \times 3$ | $5 \times 5$ | $3 \times 3$ | 12.76 | 24.47 |
|  | $3 \times 3$ | $5 \times 5$ | $5 \times 5$ | **13.20** | **24.57** |

# D EXPERIMENT SETUP

Table 13: Experiment configuration for ConvT3 and ConvS5 on long-range Moving-MNIST experiments. Overall model structure is ResNet.

| | Hyperparameters | Moving-MNIST-300 | Moving-MNIST-600 |
|---|---|---|---|
| | Params | 21M | 21M |
| | Input Resolution | $64 \times 64$ | $64 \times 64$ |
| | Latent Resolution | $16 \times 16$ | $16 \times 16$ |
| | Batch Size | 8 | 8 |
| | Sequence Length | 300 | 600 |
| | LR | $1 \times 10^{-3}$ | $1 \times 10^{-3}$ |
| | LR Schedule | cosine | cosine |
| | Warmup Steps | 5k | 5k |
| | Max Training Steps | 300K | 300K |
| | Weight Decay | $1 \times 10^{-5}$ | $1 \times 10^{-5}$ |
| Encoder | Depths | 64, 128, 256 | 64, 128, 256 |
| | Blocks | 1 | 1 |
| Decoder | Depths | 64, 128, 256 | 64, 128, 256 |
| | Blocks | 1 | 1 |
| | Hidden Dim ($U$) | 256 | 256 |
| | State Size ($P$) | 256 | 256 |
| ConvSSM | $\mathcal{B}$ Kernel Size | $3 \times 3$ | $3 \times 3$ |
| | $\mathcal{C}$ Kernel Size | $3 \times 3$ | $3 \times 3$ |
| | Layers | 8 | 8 |

Table 14: Experiment configuration for ConvT3 and ConvS5 on PDEBench experiments. Learning rate for ConvS5 on Diffusion-Reaction was reduced due to training instability.

| | Hyperparameters | Shallow-Water | Diffusion-Reaction |
|---|---|---|---|
| | Params | 6M | 6M |
| | Input Resolution | $128 \times 128$ | $128 \times 128$ |
| | Latent Resolution | $8 \times 8$ | $8 \times 8$ |
| | Batch Size | 8 | 8 |
| | Sequence Length | $16 \rightarrow 1$ | $16 \rightarrow 1$ |
| | LR | $5 \times 10^{-4}$ | $5 \times 10^{-4} / 2 \times 10^{-4}$ |
| | LR Schedule | cosine | cosine |
| | Warmup Steps | 10k | 10k |
| | Max Training Steps | 1M | 1M |
| | Weight Decay | $1 \times 10^{-5}$ | $1 \times 10^{-5}$ |
| | Hidden Dim ($U$) | 384 | 384 |
| | State Size ($P$) | 384 | 384 |
| ConvSSM | $\mathcal{B}$ Kernel Size | $1 \times 1$ | $1 \times 1$ |
| | $\mathcal{C}$ Kernel Size | $1 \times 1$ | $1 \times 1$ |
| | Layers | 6 | 6 |

Table 15: Experiment configuration for ablation studies. Overall model structure is ResNet.

| | Hyperparameters | ConvS5 | ConvT3 |
|---|---|---|---|
| | Input Resolution | $64 \times 64$ | $64 \times 64$ |
| | Latent Resolution | $16 \times 16$ | $16 \times 16$ |
| | Batch Size | 16 | 16 |
| | Sequence Length | $10 \rightarrow 10$ | $10 \rightarrow 10$ |
| | LR | $1 \times 10^{-3}$ | $1 \times 10^{-3}$ |
| | LR Schedule | cosine | cosine |
| | Warmup Epochs | 10 | 10 |
| | Max Training Epochs | 200 | 200 |
| | Weight Decay | $1 \times 10^{-5}$ | $1 \times 10^{-5}$ |
| Encoder | Depths | 64, 128, 256 | 64, 128, 256 |
| | Blocks | 1 | 1 |
| Decoder | Depths | 64, 128, 256 | 64, 128, 256 |
| | Blocks | 1 | 1 |
| | Hidden Dim ($U$) | 256 | 256 |
| | State Size ($P$) | 256 | 256 |
| | $\mathcal{B}$ Kernel Size | $3 \times 3 / 1 \times 1$ | $3 \times 3 / 1 \times 1$ |
| ConvSSM | $\mathcal{C}$ Kernel Size | $3 \times 3 / 1 \times 1$ | $3 \times 3 / 1 \times 1$ |
| | Layers | 2 / 4 / 6 / 8 / 12 | 2 / 4 / 6 / 8 / 12 |
| | Off-Diagonal Proportion | N/A | -1, -1 / 1, 1 |
| | Mini (Kernel Sharing) | N/A | Yes / No |

