# OpenReview forum: "ConvT3: Structured State Kernels for Convolutional State Space Models"
_ICLR.cc/2026/Conference — ICLR 2026 Poster_

### Official Review · Reviewer_V85B · 2025-10-20

**Soundness:** 3
**Presentation:** 3
**Contribution:** 2
**Rating:** 6
**Confidence:** 4

**Summary:**

The paper proposes an extension of the ConvSSM by enabling a larger state-space kernel size, which increases from the original value of $1$ to $3$. The model is parameterized via a tridiagonal Toeplitz tensor and asymptotic stability is guaranteed. The model is empirically validated on long-range video generation (Moving-MNIST) and physical system modeling (PDEBench) tasks.

**Strengths:**

* The derivation of the ConvT3 block is rigorous and well-grounded in tridiagonal Toeplitz matrix theory.
* The empirical results look promising, and the model seems to gain performance at a small cost of computing.
* The ablation is thorough and insightful. It demonstrates the importance of four design choices separately.

**Weaknesses:**

* It is not thoroughly analyzed in the manuscript why increasing the kernel size from $1$ to $3$ in the latent space improves the model. While a complete theory may not be very accessible without an extensive amount of work, the paper would benefit from a dedicated discussion (theoretical or empirical, but on a more insightful toy example where further analysis can be done) that illustrates the benefits of a larger convolution kernel.
* The mathematical derivation in section 3.3 is dry and purely technical. It is hard to tell what the final parameterization method is with a first glance. I would recommend that the author(s) start with a clear and accessible method and then use a theorem to show its stability property. (The proof can be dumped into an appendix.)
* While ConvT3 is as efficient as ConvS5 in the $\mathcal{O}$-notation, the constant inside is also important. This governs how the ConvT3 model is less computationally efficient in practice. It would be good to show some analysis of this kind or show some scaling plots.

**Questions:**

1. Is $P = 3$ a special number? That said, since the theory is heavily grounded in the tridiaognal Toeplitz matrix theory, I imagine that it could be hard to generalize this framework to a larger $P$ per Galois theory; is this true? How does $P = 2$ compare to $3$?
2. How does the memory usage of ConvT3 compare to that of ConvS5?
3. In Table 1, you seem to show that ConvT3 works much better than ConvS5 in the long-horizon regime (see 800 frames versus 1200 frames). Do you have a good intuition in that?

---

> ### Author Response · Authors · 2025-11-23
>
> We thank the reviewer for the positive evaluation and constructive suggestions.
>
> We are refining the manuscript to better highlight our contributions and to clarify the reasoning behind the performance improvements. Our detailed responses are provided below.
>
>
> **Weaknesses:**
>
> 1. We appreciate the reviewer’s insightful comment regarding the need for a clearer analysis or an illustrative example. We agree that a toy example would further clarify the benefit of using larger state kernels.
>
>     While a full theoretical treatment is non-trivial due to the difficulty of analytic isolation of recurrent spatial propagation, we can provide intuitive reasoning that clarifies *why* the expressive gap between 1×1 and 3×3 kernels is substantial. In ConvSSMs, the spatial propagation in the state tensors is governed solely by the state kernel $\mathbfcal{A}$. When the kernel is restricted to 1×1, the pixels of latent state evolve independently across time; no spatial mixing occurs, making the latent dynamics effectively a collection of per-pixel SSMs.
>
>     A 3×3 kernel, in contrast, introduces spatial coupling: information can flow to immediate neighbors at every recurrent step. Because this propagation compounds over long sequences, even local 3×3 coupling induces large-scale spatial interactions in the latent dynamics. This change is not just relying on more parameters, but rather a qualitative shift in the model class, enabling the representation of latent PDE-like dynamics that are fundamentally impossible under the 1×1 restriction. In this sense, the comparison is between expressible vs. inexpressible spatial behaviors. Consequently, our results emperically indicate that once this spatial propagation becomes possible, the model can capture spatiotemporal patterns more effectively across both video and PDE-bench tasks.
>
>     Following the reviewer’s suggestion, we will incorporate this discussion into the manuscript to highlight the intuition and practical implications of using larger state kernels.
>
> 2. We appreciate the reviewer’s careful reading and constructive comment. We agree that the current Section 3.3 is overly technical at first glance, which may obscure the core idea of the proposed parameterization. To improve presentation, we will restructure the section to state the high-level strategy of the Hurwitz and positivity conditioning is introduced first. Its formal proof will then follow, with detailed derivations moved to the appendix. We are also drawing an illustrative figure to clearly convey the proposed parameterization.
>
> 3. Thank you for the helpful suggestion to more explicitly address the constant factors that influence the practical efficiency of ConvT3. This point is indeed important for a fair comparison with ConvS5, and we provide both a refined complexity expression and empirical measurements to clarify this aspect.
>
>     The theoretical computational cost for ConvT3 is $\mathcal{O}(LPHWU(k_B^2 + k_C^2)+LPHW(H + W))$, which corrects the earlier expression in the Appendix. Moreover, the second term can be reduced to $\mathcal{O}(LPHWU(k_B^2 + k_C^2)+LPHW(\log H + \log W))$ by using FFT. The experimental results below were obtained without using FFT acceleration, meaning further gains are still available.
>
>     According to the reviewer's suggestion, we will include this refined complexity discussion and the scaling experiments in the revised manuscript. The following tables show the measured inference time of a single layer under various state sizes, hidden dimensions, sequence lengths, and image resolutions, highlighting that ConvT3 remains practically competitive with ConvS5 even before FFT optimization.
>
>     - Inference time under different state sizes and hidden dimensions
>
>         | State size $P$, Hidden dim $U$|	ConvS5 (Time \[ms\])	| ConvT3 (Time \[ms\]) |
>         | -------- | -------- | -------- |
>         |512|	52.7 (1.00$\times$)	|65.8 (1.25$\times$)	|
>         |256|	20.1 (1.00$\times$)	|28.0 (1.39$\times$)	|
>         |128|	10.5 (1.00$\times$)	|13.8 (1.31$\times$)	|
>         |64	| 7.2 (1.00$\times$)	|9.3 (1.29$\times$)	|
>
>     - Inference time under different sequence lengths
>
>         | Sequence length $L$|	ConvS5 (Time \[ms\])	| ConvT3 (Time \[ms\])	|
>         | -------- | -------- | -------- |
>         |100|	100.2 (1.00$\times$)	|138.6 (1.38$\times$)	|
>         |50	|50.8 (1.00$\times$)|	67.3 (1.32$\times$)	|
>         |20	|20.9 (1.00$\times$)|	27.9 (1.33$\times$)	|
>         |10	|11.9 (1.00$\times$)|	15.4 (1.29$\times$)	|
>
>     - Inference time under different image resolutions
>
>         | Image resolution $H\times W$|	ConvS5 (Time \[ms\])	| ConvT3 (Time \[ms\])	|
>         | -------- | -------- | -------- |
>         |$32\times 32$	|77.2 (1.00$\times$)	|160.6 (2.08$\times$)	|
>         |$16\times 16$	|20.9 (1.00$\times$)	|27.8 (1.33$\times$)	|
>         |$8\times 8$	|7.7 (1.00$\times$)	|9.7 (1.26$\times$)	|
>         |$4\times 4$	|7.0 (1.00$\times$)	|8.0 (1.14$\times$)	|

---

> > ### Author Response · Authors · 2025-11-23
> >
> > **Questions:**
> >
> > 1. The current ConvSSM formulation works well only for odd-sized state kernels. With symmetric padding, using an even-sized state kernel would change the output state resolution at every timestep, which would not work.
> >
> >     It is possible to think of an asymmetric setting, with the Toeplitz matrix being an upper or lower triangular matrix. This would preserve the latent image size. Then a different method for calculating the power of the matrix would be recommended, as such matrix is not directly diagonalizable. (For such asymmetric setting, calculating the power of the toeplitz matrix/tensor within constant complexity seems possible and relatively easy. Once having such method, one can follow our approach from then on, starting from the tensor product with a SSM matrix.)
> >
> >     So to directly answer the questions, yes, it is hard to generalize this framework to a larger kernel size. Toepliz matrices with larger ranks do not have closed eigendecomposition, as far as we know. It may be possible to calculate the power of Toepliz matrix/tensor without diagonalization (like the asymmetric setting), but it seems to be a matter beyond the scope of this paper. (Diagonalization is a very powerful method, as one must also consider expanding the method from 1D convolution to 2D convolution. Methods that worked on matrices may not work on multi-dimensional tensors.)
> >
> > 2. Thank you for raising this point. To address the reviewer's question, we also measured the memory usage of a single layer under identical settings. The results below show that ConvT3 exhibits memory consumption comparable to ConvS5 across a wide range of state sizes, hidden dimensions, sequence lengths, and image resolutions.
> >
> >     - Memory usage under different state sizes and hidden dimensions
> >
> >         | State size $P$, Hidden dim $U$| ConvS5 (Memory \[GB\])| 	ConvT3 (Memory \[GB\])|
> >         | -------- | -------- | -------- |
> >         |512	|4.26 (1.00$\times$)|	5.22 (1.23$\times$)|
> >         |256	|2.13 (1.00$\times$)|	2.62 (1.23$\times$)|
> >         |128|	1.09 (1.00$\times$)|	1.34 (1.22$\times$)|
> >         |64	|0.58 (1.00$\times$)|	0.70 (1.21$\times$)|
> >
> >     - Memory usage under different sequence lengths
> >
> >         | Sequence length $L$	| ConvS5 (Memory \[GB\])| 	ConvT3 (Memory \[GB\])|
> >         | -------- | -------- | -------- |
> >         |100|	10.46 (1.00$\times$)	|12.39 (1.18$\times$)|
> >         |50	|5.25 (1.00$\times$)|	6.28 (1.19$\times$)|
> >         |20	|2.13 (1.00$\times$)|	2.62 (1.22$\times$)|
> >         |10	|1.09 (1.00$\times$)|	1.40 (1.28$\times$)|
> >
> >     - Memory usage under different image resolutions
> >
> >         | Image resolution $H\times W$| ConvS5 (Memory \[GB\])| 	ConvT3 (Memory \[GB\])|
> >         | -------- | -------- | -------- |
> >         |$32\times 32$	|8.25 (1.00$\times$)	|11.75 (1.42$\times$)|
> >         |$16\times 16$	|2.13 (1.00$\times$)	|2.62 (1.23$\times$)|
> >         |$8\times 8$	|0.60 (1.00$\times$)	|0.70 (1.16$\times$)|
> >         |$4\times 4$	|0.22 (1.00$\times$)	|0.24 (1.08$\times$)|
> >
> >
> > 3. We conjecture the widening performance gap in the long-horizon regime stems from the difference in how spatial information propagates through the latent state. ConvS5 relies on a 1×1 state kernel, meaning its temporal evolution mixes information only channel-wise and cannot transport any spatial structure across time. As prediction horizon grows, this limited propagation accumulates error and eventually leads to drift.
> >
> >     ConvT3, in contrast, uses 3×3 state kernels that enable local spatial mixing in the latent state at every step. This allows the model to maintain and update coherent spatial patterns throughout the recurrent rollout, which becomes increasingly important as the horizon becomes longer. As a result, the benefits of extended spatial propagation become more pronounced at 1200 frames, where long-range consistency dominates overall performance.

---

> > > ### Comment · Reviewer_V85B · 2025-11-28
> > >
> > > Thank you for your thorough rebuttal. As Reviewer pYFy pointed out, it is useful to scale up the resolution to see if the ratio eventually converges to a small number. Thank you also for clarifying the difficulty with other kernel sizes: it is good to have this discussion in the paper. I believe this is a good contribution and keep my positive evaluation.

---

### Official Review · Reviewer_a6wG · 2025-10-28

**Soundness:** 3
**Presentation:** 2
**Contribution:** 2
**Rating:** 4
**Confidence:** 2

**Summary:**

This paper proposes a ConvSSM method using Tridiagonal Toeplitz Tensors, which equivalently implements a ConvSSM with 3 $\times$ 3 state convolution. Compared with traditional ConvSSM, this method give a constrained tridiagonal Toeplitz tensor. This method can avoid exploding computation in parallel scans with larger kernel and learn state dynamics from effectively capturing spatiotemporal context.

**Strengths:**

1. This paper proposes a new method which are used in statistical theory and leverage this method into state space model.
2. This paper gives the proofs in detail and gives some theorems and definitions.
3. The experiments prove that this method is good enough compared traditional methods on video generation tasks and complex physical system modeling.

**Weaknesses:**

1. This paper introduces an excessive number of notations, with some lacking sufficient detail—for instance, the formula in line 196 is not adequately elaborated.
2. The paper fails to theoretically validate the effectiveness of ConvT3 in comparison to ConvSSM. It is suggested that the authors supplement a dedicated theoretical analysis section, rather than relying solely on experimental results.
3. Regarding the experimental validation, only two experiments are presented. However, classification and detection tasks are standard benchmarks in computer vision. Additionally, traditional SSM-based methods (e.g., Vmamba) have been applied to CV tasks. It is recommended to include more comparative experiments covering these typical tasks.
4. Certain theoretical details require clarification. Specifically, the calculation process of Equation 9 and its underlying rationale are not sufficiently explained.
5. It would be valuable to provide a comparison of time complexity/cost between the proposed method and transformer-based approaches.

**Questions:**

See weakness

---

> ### Author Response · Authors · 2025-11-23
>
> We appreciate the reviewer for recognizing both the theoretical and empirical contributions of our work.
> To clarify, the proposed tridiagonal Toeplitz (TT) tensor enables ConvSSMs with 3×3 state convolutions, maintaining linear-time scans and stable training while expanding the expressive capacity of ConvSSMs with 1x1 state convolutions.
>
> We thoroughly addressed reviewer’s comments on presentation and additional analysis as follows.
>
> >1. This paper introduces an excessive number of notations, with some lacking sufficient detail-for instance, the formula in line 196 is not adequately elaborated.
>
>
> - Thank you for this very helpful comment. We agree that the manuscript introduces many notations without sufficient guidance. While we provided a consolidated notation table in the Appendix, we realize that the main text does not clearly direct the reader to it. Including this, we are revising the presentation to make it more streamlined and easier to follow.
>
> - Moreover, the expression at line 196 uses a Python-style indexing notation, which is a non-standard mathmetical representation. We understand that this requires clearer explanation, and we will add an explanatory remark to avoid confusion in the revised version that we will upload before the rebuttal period ends.
>
> >2. The paper fails to theoretically validate the effectiveness of ConvT3 in comparison to ConvSSM. It is suggested that the authors supplement a dedicated theoretical analysis section, rather than relying solely on experimental results.
>
>
> - Thank you for the comment on the theoretical validation. To avoid misunderstanding, both ConvS5 and ConvT3 belong to the ConvSSM family, which is defined for general state kernel sizes; the key distinction is that ConvS5 has been limited to ConvSSM with 1×1 state kernels, whereas our method, ConvT3 makes 3×3 state kernels feasible while retaining the same efficiency properties.
>
> - By introducing a structured Toeplitz parameterization, we expands the implementable class of ConvSSMs. In particular, Section 3 provides the theoretical construction showing how 3×3 state propagation can be realized while preserving the stability and efficiency, and Section 5 confirms these design through empirical evaluation. Therefore, we believe the manuscript already provides both the theoretical justification and practical validation necessary for the proposed extension.
>
> > 3. Regarding the experimental validation, only two experiments are presented. However, classification and detection tasks are standard benchmarks in computer vision. Additionally, traditional SSM-based methods (e.g., Vmamba) have been applied to CV tasks. It is recommended to include more comparative experiments covering these typical tasks.
>
>
> - Thank you for the comment on experimental validation. To clarify, our work is motivated by spatiotemporal modeling to capture coupled spatial–temporal dynamics. Tasks like video prediction or physical simulations provide a more direct and meaningful evaluation of the temporal propagation mechanisms that our 3×3 state kernel design aims to enhance. This is also consistent with prior research, including ConvS5 and MPP, which  evaluate their models on video prediction and other spatiotemporal forecasting tasks, as these tasks best reflect the modeling capabilities such architectures seek to improve. Classification and detection tasks, in contrast, are heavily influenced by semantic recognition and object-level reasoning rather than by a model’s capacity to propagate and represent spatiotemporal dynamics.
>
> - For this reason, our experiments focus on domains where temporal dynamics play a central role. In particular, PDEBench serves as a primary benchmark for physical simulation dynamics, and we include two distinct datasets within PDEBench to validate the method under different types of spatiotemporal behavior. We believe these experiments offer an appropriate and thorough evaluation aligned with the core objective of our model design.
>
> > 4. Certain theoretical details require clarification. Specifically, the calculation process of Equation 9 and its underlying rationale are not sufficiently explained.
>
> - Thank you for pointing this out. Equation (9) is simply the expanded form of the zero-padded convolution definition; it does not involve any additional derivation beyond the standard convolution operation itself. We agree, however, that our current presentation requires clarification, and we are revising the manuscript to make the rationale and notation clearer. If there are any similar points of confusion, we would be happy to address them in the revised manuscript.

---

> > ### Author Response · Authors · 2025-11-23
> >
> > > 5. It would be valuable to provide a comparison of time complexity/cost between the proposed method and transformer-based approaches.
> >
> >
> > - Thank you for the suggestion. The theoretical computational cost for ConvT3 is $\mathcal{O}(LPHWU(k_B^2 + k_C^2)+LPHW(H + W))$, which corrects the earlier expression in the Appendix. (The second term can be reduced to $\mathcal{O}(LPHWU(k_B^2 + k_C^2)+LPHW(\log H + \log W))$ by using FFT.)
> >
> > - As shown here, ConvT3 retains the *linear-time* complexity in the sequence length $L$, inherited from ConvSSMs’ parallel scan formulation. In contrast, transformer-based approaches generally incur *quadratic* complexity in $L$ due to the self-attention operation, or linear complexity only when additional approximation mechanisms (e.g., kernelization or sparsification) are introduced. This highlights a fundamental difference in scalability with respect to sequence length. We will clarify this comparison more explicitly in the revised manuscript.

---

### Official Review · Reviewer_pYFy · 2025-10-30

**Soundness:** 2
**Presentation:** 3
**Contribution:** 3
**Rating:** 6
**Confidence:** 2

**Summary:**

The paper proposes ConvT3, a convolutional state-space model (ConvSSM) that realizes a 3×3 state kernel while keeping linear-time parallel scans for long sequences. The key idea is to structure the state kernel as (i) a diagonalizable SSM matrix over hidden channels and (ii) a tridiagonal Toeplitz (TT) tensor over spatial axes. This structure enables efficient training while capturing richer spatial dynamics than prior 1×1 state kernels (e.g., ConvS5). On Moving-MNIST, ConvT3 achieves state-of-the-art results across most metrics; on PDEBench, it attains the best accuracy with efficiency close to ConvS5.

**Strengths:**

+ The paper shows how to move from 1×1 to 3×3 state kernels in ConvSSMs without losing linear-time scans, via a structured (diagonalizable) state tensor and tridiagonal Toeplitz (TT) formulation.

+ Training stability: The paper uses a Hurwitz condition-based parameterization (constraining eigenvalues to have negative real parts) and a positive eigen-tensor construction to keep dynamics stable during training. This is validated by the training loss curve (Figure 3).

+ Ablations on kernel size and minimal parameterization: MiniT3 outperformed ConvS5 despite the minimal increase in parameters, and the kernel-size ablations show the 3×3 state kernel A is the main driver of gains, not B/C alone.

**Weaknesses:**

- Limited benchmarks: Moving-MNIST and PDEBench tasks show the benefits, but they are relatively synthetic. Adding natural video datasets or harder physics targets (e.g., Navier–Stokes) would strengthen the paper.

- Compute and throughput details: The paper states linear-time scans and that efficiency is close to ConvS5, but the actual comparisons are limited. Per-component speed and throughput are missing. Also, efficiency measurement across sequence lengths and resolutions would make the efficiency claims stronger.

- Ablation: The kernel-size studies of A, B, and C are helpful, but additional ablations such as 3×3 vs 5×5 state kernels would be beneficial.

**Questions:**

See the weaknesses.

---

> ### Author Response · Authors · 2025-11-23
>
> We appreciate the reviewer’s careful reading and understanding of the main idea of our work.
>
> In response to the reviewer’s concerns about the empirical validation, we present our clarification along with additional experimental results below.
>
> 1. We respect the reviewer’s point regarding the limited realism of validation tasks. We would like to note that Moving-MNIST is entirely synthetic, whereas PDEBench consists of physics-based simulation data that reflects realistic dynamics.
>
>     Moreover, the primary goal of this work is to introduce ConvT3 by extending the state kernel of ConvS5 to a structured 3×3 formulation. In this context, the key question we examine is whether this kernel extension improves modeling capability within the ConvSSM framework. Spanning multiple tasks and supported by ablation studies, the experiments presented in the paper consistently show that ConvT3 outperforms ConvS5, providing strong evidence for the effectiveness and generality of the proposed kernel design.
>
>     For these reasons, we believe that evaluations on additional tasks, while meaningful, would largely reaffirm the findings already demonstrated by our current results.
>
>
> 2. We appreciate the reviewer’s comment regarding the limited compute and throughput analysis. To address this point, we conducted additional measurements comparing a *single ConvSSM layer* of ConvT3 and ConvS5 under various model and input configurations.
>
>     The theoretical computational cost for ConvT3 is $\mathcal{O}(LPHWU(k_B^2 + k_C^2)+LPHW(H + W))$, which corrects the earlier expression in the Appendix. Moreover, the second term can be reduced to $\mathcal{O}(LPHWU(k_B^2 + k_C^2)+LPHW(\log H + \log W))$ by using FFT. The experimental results below were obtained without using FFT acceleration, meaning further gains are still available.
>
>     According to the reviewer's suggestion, we will include this efficiency discussion and the scaling experiments in the revised manuscript. The following tables show the measured inference time and memory usage of a single layer under various state sizes, hidden dimensions, sequence lengths, and image resolutions, highlighting that ConvT3 remains practically competitive with ConvS5 even before FFT optimization, as well as maintaining the linear-time scalability in sequence lengths.
>
>     - Inference time and memory usage under different state sizes and hidden dimensions
>
>         | State size $P$, Hidden dim $U$|	ConvS5 (Time \[ms\])	| ConvT3 (Time \[ms\])	| ConvS5 (Memory \[GB\])| 	ConvT3 (Memory \[GB\])|
>         | -------- | -------- | -------- | -------- | -------- |
>         |512|	52.7 (1.00$\times$)	|65.8 (1.25$\times$)	|4.26 (1.00$\times$)|	5.22 (1.23$\times$)|
>         |256|	20.1 (1.00$\times$)	|28.0 (1.39$\times$)	|2.13 (1.00$\times$)|	2.62 (1.23$\times$)|
>         |128|	10.5 (1.00$\times$)	|13.8 (1.31$\times$)	|1.09 (1.00$\times$)|	1.34 (1.22$\times$)|
>         |64	| 7.2 (1.00$\times$)	|9.3 (1.29$\times$)	|0.58 (1.00$\times$)|	0.70 (1.21$\times$)|
>
>     - Inference time and memory usage under different sequence lengths
>
>         | Sequence length $L$|	ConvS5 (Time \[ms\])	| ConvT3 (Time \[ms\])	| ConvS5 (Memory \[GB\])| 	ConvT3 (Memory \[GB\])|
>         | -------- | -------- | -------- | -------- | -------- |
>         |100|	100.2 (1.00$\times$)	|138.6 (1.38$\times$)	|10.46 (1.00$\times$)	|12.39 (1.18$\times$)|
>         |50	|50.8 (1.00$\times$)|	67.3 (1.32$\times$)	|5.25 (1.00$\times$)|	6.28 (1.19$\times$)|
>         |20	|20.9 (1.00$\times$)|	27.9 (1.33$\times$)	|2.13 (1.00$\times$)|	2.62 (1.22$\times$)|
>         |10	|11.9 (1.00$\times$)|	15.4 (1.29$\times$)	|1.09 (1.00$\times$)|	1.40 (1.28$\times$)|
>
>     - Inference time and memory usage under different image resolutions
>
>         | Image resolution $H\times W$|	ConvS5 (Time \[ms\])	| ConvT3 (Time \[ms\])	| ConvS5 (Memory \[GB\])| 	ConvT3 (Memory \[GB\])|
>         | -------- | -------- | -------- | -------- | -------- |
>         |$32\times 32$	|77.2 (1.00$\times$)	|160.6 (2.08$\times$)	|8.25 (1.00$\times$)	|11.75 (1.42$\times$)|
>         |$16\times 16$	|20.9 (1.00$\times$)	|27.8 (1.33$\times$)	|2.13 (1.00$\times$)	|2.62 (1.23$\times$)|
>         |$8\times 8$	|7.7 (1.00$\times$)	|9.7 (1.26$\times$)	|0.60 (1.00$\times$)	|0.70 (1.16$\times$)|
>         |$4\times 4$	|7.0 (1.00$\times$)	|8.0 (1.14$\times$)	|0.22 (1.00$\times$)	|0.24 (1.08$\times$)|

---

> > ### Author Response · Authors · 2025-11-23
> >
> > 3. We have conducted additional experiments using 5x5 $\mathbfcal{B}$ and $\mathbfcal{C}$ kernels, which correspond to a new row added to the original Table 6. We note that extending the $\mathbfcal{A}$ kernel to 5x5 falls outside the scope of this work, and such experiments are not feasible within our current formulation.
> >
> >     Consistent with the previous results, ConvT3 with a 3×3 $\mathbfcal{A}$ kernel outperforms ConvS5, which uses 1×1 $\mathbfcal{A}$ kernels with the same-sized $\mathbfcal{B}$ and $\mathbfcal{C}$ kernels, in most cases. These results support our main claim regarding the advantage of incorporating structured spatial–temporal propagation via the 3×3 state kernel. Also note that the best performing model in this ablation study is a ConvT3 model.
> >
> >     We will update the manuscript with an extended version of Table 6 to include the new results.
> >
> >     - (Additional rows of Table 6) (**Ablations: Kernel size.**) The introduction of a 3×3 state kernel $\mathbfcal{A}$ resulted in notable performance gains. Bold: better performance under same-sized $\mathbfcal{B}$ and $\mathbfcal{C}$ kernels.
> >
> >         | Model | $\mathbfcal{A}$ kernel | $\mathbfcal{B}$ kernel | $\mathbfcal{C}$ kernel | MSE $\downarrow$ | MAE $\downarrow$ |
> >         | -------- | -------- | -------- | -------- | -------- | -------- |
> >         | ConvS5   |$1\times 1$ |	$3\times 3$	|$5\times 5$|	11.49|  	22.83|
> >         |  | $1\times 1$|	$5\times 5$|	$3\times 3$|	**12.48** | 	**24.43**|
> >         |  | $1\times 1$|	$5\times 5$|	$5\times 5$|	13.56  |	24.73|
> >         |  | $1\times 1$|	$1\times 1$|	$5\times 5$|	10.56  |	21.03
> >         |  | $1\times 1$|	$5\times 5$|	$1\times 1$|	11.82  |	23.52|
> >         |ConvT3	|$3\times 3$	|$3\times 3$	|$5\times 5$	|**11.34** | 	**22.43**|
> >         |       |$3\times 3$	|$5\times 5$	|$3\times 3$	|12.76  |	24.47|
> >         |       |$3\times 3$	|$5\times 5$	|$5\times 5$	|**13.20**  |	**24.57**|
> >         |       |$3\times 3$    |$1\times 1$    |$5\times 5$    |**9.91**  |	**20.38**|
> >         |       |$3\times 3$	|$5\times 5$	|$1\times 1$	|**11.48**  |	**23.09**|

---

> ### Comment · Reviewer_pYFy · 2025-11-24
>
> Thank you for the additional tables.
> From the Inference time comparison table, I notice that the speed increase is substantial at higher input resolutions, which might become a bottleneck during model scaling. Are there any possible approaches to address this issue?
> Additionally, in the last table, the 1×1 kernel for B and the 5×5 kernel for C show noticeably better performance compared to other configurations. Could you elaborate on the reason behind this improvement?

---

> > ### Author Response · Authors · 2025-11-28
> >
> > >From the Inference time comparison table, I notice that the speed increase is substantial at higher input resolutions, which might become a bottleneck during model scaling. Are there any possible approaches to address this issue?
> >
> > - Thank you for raising this practical concern regarding inference speed at higher input resolutions. We see two complementary approaches to alleviate this issue.
> >     1. The ConvT3 implementation can be accelerated by replacing the explicit multiplication with the eigenbasis matrix $Q$ of the tridiagonal Toeplitz operator with FFT. The multiplication by $Q$ corresponds exactly to a type-I discrete sine transform (DST), so it can be implemented via FFT-based routines instead of dense matrix multiplication. While the direct matrix multiplication has quadratic complexity in the spatial resolution (e.g., $\mathcal{O}(H^2)$ with respect to height $H$), the FFT-based implementation reduces this to $\mathcal{O}(H \log H)$. In practice, this means that, relative to ConvS5, the extra cost scales only logarithmically with the resolution, rather than linearly as in the current implementation.
> >     2. Since the effective cost of ConvT3 depends on the hidden feature resolution rather than raw input resolution, the input can first be encoded into a lower-resolution latent feature using an encoder (e.g., via strided convolutions or patch embeddings). Applying ConvT3 to these feature allows the model to exploit larger state kernels without incurring prohibitive cost at very high input resolutions. By designing the encoder to map inputs to a more moderate hidden resolution, the ConvT3 layer can be used without becoming a major bottleneck.
> >
> > >Thank you for the additional tables. From the Inference time comparison table, I notice that the speed increase is substantial at higher input resolutions, which might become a bottleneck during model scaling. Are there any possible approaches to address this issue? Additionally, in the last table, the 1×1 kernel for B and the 5×5 kernel for C show noticeably better performance compared to other configurations. Could you elaborate on the reason behind this improvement?
> >
> > - Thank you for this insightful observation. As shown in the table, configurations with larger output kernel $\mathbfcal{C}$ generally outperform those with smaller $\mathbfcal{C}$, whereas enlarging the input kernel $\mathbfcal{B}$ beyond 1x1 does not yield consistent gains and can even degrade performance. This pattern appears in both ConvS5 and ConvT3. Specifically, smaller $\mathbfcal{B}$ seems to perform better as the $\mathbfcal{C}$ kernel gets larger. This suggests that the input kernel $\mathbfcal{B}$ is more sensitive to overfitting.
> > - One hypothesis is that when $\mathbfcal{B}$ is spatial (3x3 or 5x5), it introduces early spatial mixing and can blur local input patterns before the recurrent dynamics act, which explains the degradation observed for larger $\mathbfcal{B}$. A 1x1 kernel for $\mathbfcal{B}$ instead acts as a pure channel-wise projection, preserving local structure and letting $\mathbfcal{A}$ (3×3 in ConvT3) and $\mathbfcal{C}$ (5×5) handle spatial reasoning and readout. This division of roles leads to the best-performing configuration with $\mathbfcal{B}$=1x1 and $\mathbfcal{C}$=5x5.
> > - Importantly, these findings are fully consistent with our main claim: the performance improvement mainly stems from enhanced spatial propagation in the latent state through the state kernel $\mathbfcal{A}$, which ConvT3 enables while preserving the linear-time parallel-scan property. We appreciate the reviewer’s careful analysis, which helped us clarify this behavior in the revision.

---

### Comment · Area_Chair_7N7w · 2025-11-24
**Discussion with Authors**

Dear Reviewers,

The authors have diligently provided responses to your questions and concerns. I request you to please review the authors' responses, acknowledge that you have read them and actively engage with them in further discussion as needed.

This discussion period, with the authors, will end on December 2, 2025 (AoE). However, I request that you not wait until the last minute and actively engage with the authors early.

Best,
AC

---

### Author Response · Authors · 2025-12-03
**Revision Summary**

Dear Reviewers and AC,

We appreciate all the careful comments on our work. We have revised the initial manuscript to reflect the discussions we had with the reviewers.

A summary of the reivisions is provided below:

**Addressing Reviewer pYFy's comment (Ablation study)**
- Section 5, Line 487
- Appendix C, New Table 12 (Lines 1063--1075)

**Addressing Reviewer a6wG and V85B's comments (Readability)**
- Section 2, Lines 75--76
- Section 3, Lines 261--327; New Figure 3 (Lines 270--282)

**Addressing Reviewers pYFy, a6wG, V85B's comments (Computational cost)**
- Appendix B, Lines 918--968; New Tables 8--10 (Lines 943--968)

We believe that the revised manuscript now provides the contributions and validations of our proposed method more clearly. Thank you again for all the helpful feedback.

---

### Meta-Review · Area_Chair_BBG6 · 2026-01-05

**Summary:**

Modeling spatio-temporal sequences is an important yet challenging problem in computer vision and scientific applications. This paper introduces an interesting and principled approach that addresses limitations of the original Convolutional State Space Model (ConvSSM). The authors propose a new ConvSSM formulation that leverages Tridiagonal Toeplitz Tensors. The method is well-motivated, and the empirical results demonstrate clear performance advantages.

Overall, reviewer sentiment is very positive. The rebuttal was used effectively to address the main concerns, and in particular, the authors responded well to the issues raised by reviewer a6wG. The paper makes a solid contribution. As noted by several reviewers, the presentation could be improved by streamlining notation and moving some technical details to the appendix to improve accessibility.

In summary, this is a nice and interesting paper that is applicable to a wide range of applications. Thus, I recommend accept.

**Reviewer Concerns:**

The rebuttal effectively addressed all major concerns raised by the reviewers.

**Reviewer Scores:**

It seems likely that V85B and pYFy would have maintained their positive scores, while a6wG (low confidence) may have increased their score in light of the rebuttal.

---

### Decision · Program_Chairs · 2026-01-26

Accept (Poster)